# Refining Remote Sensing precipitation Datasets in the South Pacific with An Adaptive Multi-Method Calibration Approach

Óscar Mirones[1], Joaquín Bedia[2,3], Sixto Herrera[2], Maialen Iturbide[1], and Jorge Baño Medina[1,4]

[1]Santander Meteorology Group. Instituto de Física de Cantabria IFCA, CSIC-UC, 39005 Santander, Spain
[2]Dept. Matemática Aplicada y Ciencias de la Computación (MACC), Universidad de Cantabria, 39005 Santander, Spain
[3]Data Science and Climate Group, Universidad de Cantabria, Unidad Asociada al CSIC (Spain)
[4]Center for Western Weather and Water Extremes, Scripps Institution of Oceanography, University of California, San Diego, La Jolla, California, USA

**Correspondence:** J. Bedia (bediaj@unican.es)

**Abstract.**

Calibration techniques refine numerical model outputs for climate research, often preferred for their simplicity and suitability in many climate impact applications. Atmospheric pattern classifications for conditioned transfer function calibration, common in climate studies, are seldom explored for satellite product calibration, where significant biases may occur compared to in situ meteorological observations. This study proposes a new 'adaptive' calibration approach, applied to the Tropical Rainfall Measuring Mission (TRMM) precipitation product across multiple stations in the South Pacific. The methodology involves the daily classification of the target series into five distinct Weather Types (WTs) capturing the diverse spatio-temporal precipitation patterns in the region. Various quantile mapping (QM) techniques, including empirical (eQM), parametric (pQM), and Generalized Pareto Distribution (gpQM), as well as an ordinary scaling, are applied to each WT. We perform a comprehensive validation by evaluating 10 specific precipitation-related indices that hold significance in impact studies, which are then combined into a single Ranking Framework (RF) score, which offers a comprehensive evaluation of the performance of each calibration method for every Weather Type. These indices are assigned user-defined weights, allowing for a customized assessment of their relative importance to the overall RF score. Thus, the adaptive approach selects the best performing method for each WT based on the RF score, yielding an optimally calibrated series.

Our findings indicate that the adaptive calibration methodology surpasses standard and weather-type conditioned methods based on a single technique, yielding more accurate calibrated series in terms of mean and extreme precipitation indices, consistently across locations. Moreover, this methodology provides the flexibility to customize the calibration process based on user preferences, thereby allowing for specific indices, such as extreme rainfall indicators, to be assigned higher weights. This ability enables the calibration to effectively address the influence of intense rainfall events on the overall distribution. Furthermore, the proposed adaptive method is highly versatile and can be applied to different scenarios, datasets, and regions, provided that a prior weather typing exists to capture the pertinent processes related to regional precipitation patterns. Open-source code and illustrative examples are freely accessible to facilitate the application of the method.

**Keywords** – weather types, quantile mapping, extreme precipitation, precipitation indices, bias adjustment, TRMM

# 1 Introduction

Satellite rainfall products serve as crucial sources of information for various hydrological applications, offering continuous temporal coverage and consistent spatial estimates of precipitation in regions lacking sufficient rain gauge data. However, unlike direct observations, satellite measurements are prone to systematic errors originating from uncertainties in estimating precipitation amounts from radar reflectivity measurements (Simpson et al., 1996; Sekaranom and Masunaga, 2019) or the irregular timing of satellite overpass (Aghakouchak et al., 2009), among others. Consequently, these products often deviate

significantly from the statistical properties of observed series, particularly concerning extreme precipitation events (Mirones et al., 2023), thus requiring calibration before their application in impact studies.

In particular, the Tropical Rainfall Measuring Mission (TRMM) was a research satellite launched by NASA in 1997 to improve the understanding of the distribution and variability of precipitation in the tropics and subtropics. TRMM used several space-borne instruments to measure rainfall and its associated heat release for the first time, essential for regulating Earth's

climate. TRMM ended its operation in 2015, after providing precipitation data for 17 years, with a derived daily accumulated precipitation product (TRMM Multi-satellite Precipitation Analysis, TMPA) that have been discontinued as of Dec. 31, 2019 (Huffman et al., 2016). To continue and extend the legacy of TRMM, NASA developed the Integrated Multi-satellitE Retrievals for GPM (IMERG) algorithm, which combines information from the GPM satellite constellation and other sources to estimate precipitation over most of the globe in a very flexible framework (Huffman et al., 2020). IMERG also incorporates TRMM-era

data, creating a continuous precipitation dataset spanning over two decades. In this study, we focus on TRMM due to this long history of usage. TRMM had a low-altitude (402.5 km), non-sun-synchronous orbit, which allowed it to sample the diurnal cycle of precipitation and capture the variability of rainfall at different times of the day (National Research Council, 2006). IMERG, on the other hand, uses data from multiple satellites with different orbits and sensors, which may introduce uncertainties and inconsistencies in the diurnal sampling (Zhou et al., 2023). Thus, although new products have emerged that offer

better performance than TRMM for many hydrological applications, TRMM still has some advantages in specific contexts. In addition, we have found that comparable biases are still present in the newer IMERG product for most locations (Appendix C), thus requiring also calibration (see e.g. Arshad et al., 2021). Therefore, to illustrate the adaptive calibration methodology, the choice of data source is of secondary importance. The focus of our study is on the calibration methodology itself. Additionally, the proposed methodology, which calibrates data based on regional-scale atmospheric processes and validates the results, is

applicable to any dataset.

Essentially, the calibration process entails adjusting a transfer function that relates the parameters of raw satellite precipitation distribution to observed rain gauge time series. The effectiveness of bias reduction through post-processing depends on the underlying mechanisms producing the bias (see e.g. Maraun et al., 2017), as well as the appropriateness and accurate implementation of the chosen technique. Moreover, it is crucial to accompany this process with a proper estimation of the

associated uncertainty. In particular, the TRMM biases are not constant but associated with specific meteorological conditions, often exhibiting a systematic overestimation during wet periods and underestimation during dry periods in some regions (Islam et al., 2010; Almazroui, 2011; As-syakur et al., 2016; Giarno et al., 2018). Hence, it is reasonable to anticipate that incorporat-

ing explicit information regarding the synoptic-scale meteorological conditions into the calibration process would enhance the fitting of the transfer function. In this context, weather typing techniques (see Huth et al., 2008, for a comprehensive review), prove helpful in defining relevant weather patterns by summarizing distinct atmospheric configurations associated with different precipitation regimes (Baltaci et al., 2015; Hay et al., 1991; Riediger and Gratzki, 2014; Trigo and DaCamara, 2000). This approach can effectively situate the calibration within the context of significant atmospheric circulation processes that impact the target variable (Mirones et al., 2023), considering that the biases may be different depending on the prevailing atmospheric processes at each moment (Jury et al., 2019), so that a generalist adjustment may not allow to solve them efficiently in all cases. Moreover, although conditioning reduces the sample size, it has been shown that the calibration with adequate sub-samples can significantly enhance the reliability of the corrected series, as shown with some popular calibration methods like quantile mapping (Reiter et al., 2018).

Atmospheric pattern classifications for conditioned transfer function calibration have been already used in statistical downscaling for climate change studies (Stehlik and Bardossy, 2002; Wetterhall et al., 2007, 2012) as well as in seasonal forecasting applications (Manzanas and Gutiérrez, 2019) and short-term forecast calibration refinement (Vuillaume and Herath, 2017). However, they have rarely been investigated in the context of satellite product calibration. In a recent study, Mirones et al. (2023) proposed a novel approach for calibrating TRMM data in the South Pacific region. The methodology incorporates scaling and empirical quantile mapping techniques, conditioned to the dominant modes of interannual variability captured by specific precipitation types. This region encompasses the South Pacific Convergence Zone (SPCZ), characterized by a distinct band of low-level convergence and enhanced cloudiness extending across the South Pacific (Australian Bureau of Meteorology and CSIRO, 2011). The SPCZ is associated with notable meteorological phenomena such as heavy rainfall, convective storms, and the displacement of the intertropical convergence zone (ITCZ, Waliser and Gautier, 1993). The defined weather types were thus designed to capture the key characteristics of the regional precipitation regime while ensuring a sufficient sample size for robust conditional model fitting. Building upon this methodological framework, this study aims to further explore the potential of conditioned calibration for improving the quality of TRMM precipitation data. As a result, the new calibration method presented relies on a weather type classification designed for the synoptic characterization of regional precipitation. In this study, we built upon an already constructed classification for the study area (Mirones et al., 2023); its applicability to new regions is therefore constrained by a previous weather typing able to capture the main regional features. We expand the range of calibration techniques by incorporating a broader selection of commonly used parametric and non-parametric methods, including linear scaling, empirical quantile mapping (eQM), parametric quantile mapping (pQM), and generalized Pareto Distribution quantile mapping (gpQM), the latter adapted for a more specific treatment of extreme values in the quantile adjustment. Moreover, we examine the feasibility of combining different calibration techniques for the same location, considering the various weather types and applying a specific statistical correction method for each WT individually. Then, we assess the performance of these calibration techniques by using user-defined validation indices and suitable validation measures. The validation results for each index are globally evaluated using a weighted Ranking Framework score, which allows us to identify the optimal combination of techniques for site-based calibration.

## 2 Data and methods

### 2.1 Data

The reference observations used as the predictand for calibration were obtained from the Pacific Rainfall Database (PACRAIN, Greene et al., 2008). The PACRAIN Database comprises daily and monthly rainfall records from a comprehensive collection of rain gauge stations situated across atolls and islands in the South Pacific region. These records are sourced from various institutions, including the National Institute of Water and Atmospheric Research of New Zealand (NIWA, www.niwa.cri.nz), the US National Centers for Environmental Information (NCEI, https://www.ncei.noaa.gov/), the French Polynesian Meteorological Service (https://meteo.pf), the Schools of the Pacific Rainfall Climate Experiment (SPaRCE, https://sparce.ou.edu), and the Atlas of Pacific Rainfall (Taylor, 1973). Despite the seemingly ample raw samples within the database, an examination of missing data reveals a significantly reduced number of suitable data points. Two critical considerations arise in this context. i) Bias Correction Requirements: achieving robust fits for the various statistical methods employed in bias correction demands a relatively large sample size. This is especially true for effectively characterizing extreme events, a main point in our study due to their paramount importance in numerous hydrological applications. ii) Representativity of Locations: The chosen locations must encompass a representative spectrum of variability within the region. In our study, these locations are strategically distributed across the entire domain, offering a sensible representation of diverse precipitation regimes across the weather typing spatial domain.

As a result, we used the final subset of suitable rain gauge stations presented in Table 1. The calibrated dataset in this study is the Tropical Rainfall Measuring Mission 3B42_7B Daily product (TRMM TMPA Precipitation L3 1 day $0.25 \times 0.25$ degree V7, Huffman et al., 2016)[1]. This dataset provides measurements of daily accumulated precipitation, covering the period from January 1 1998, to January 1 2020, with a daily temporal resolution. The TRMM 3B42_7B is a research-grade dataset designed for climatological and hydrological analyses, which combines satellite-based precipitation estimates with ground-based gauge data to enhance accuracy. Specifically, it applies bias corrections using monthly gauge analysis from the Global Precipitation Climatology Centre (GPCC, Schneider et al., 2011), which helps reduce systematic errors present in the raw satellite estimates. The spatial coverage of the dataset ranges from $50.0°N$ to $50.0°S$ and $180.0°E$ to $180.0°W$. For the calibration process, the TRMM data were extracted at the nearest grid point to each rain gauge location.

### 2.2 Weather typing

We used Princial Component Analysis (PCA) to obtain representative precipitation patterns and then performed a clustering approach. We chose $K$-means, a traditional method that divides the feature space into a fixed ($K$) number of clusters by iteratively finding group centroids that maximize cluster distances and minimize within-cluster dispersion (e.g. Pike and Lintner, 2020).

---

[1]https://disc.gsfc.nasa.gov/datasets/TRMM_3B42_Daily_7/summary

**Table 1.** Final set of rain gauge stations from the PACRAIN database used in this study. The columns provide information such as the PACRAIN ID, indicating the data source (NZ for NIWA, US for NCEI, and SP for SPaRCE), station name and location, longitude and latitude coordinates in degrees, time coverage of the time series (start and end dates; an asterisk indicates data outside the TRMM period, which were discarded in this study), percentage of missing data within the start-end period, and elevation in meters above sea level.

| Station ID | Station Name | Longitude | Latitude | Start | End | % Missing Data | Altitude |
|---|---|---|---|---|---|---|---|
| NZ75400 | Kolopelu (Wallis and Futuna) | 178.12 ° W | 14.32° S | 1998-01-01 | 2012-01-01 | 9.74 | 36 |
| NZ82400 | Alofi (Niue) | 169.93° W | 19.07° S | 1998-01-01* | 2010-09-02 | 2.68 | 59 |
| NZ84317 | Rarotonga (Cook Islands) | 159.80° W | 21.20° S | 1999-09-28 | 2012-01-02 | 11.36 | 4 |
| NZ99701 | Raoul Island (New Zealand) | 177.93° W | 29.23° S | 1998-01-01* | 2012-01-01 | 0.72 | 49 |
| SP00646 | Port Vila (Vanuatu) | 168.30° E | 17.72° S | 2000-01-26 | 2013-06-01 | 18.13 | 24 |
| US14000 | Aoloau (American Samoa) | 170.77° W | 14.30° S | 1998-01-01* | 2019-12-31* | 21.72 | 408 |
| US14690 | Nu'uuli (American Samoa) | 170.70° W | 14.32° S | 1998-01-01* | 2019-12-31* | 0.037 | 3 |

The variables used for clustering were precipitation, sea-level pressure, the day-to-day sea-level pressure difference and eastward and northward wind components, all extracted from the ERA5 reanalysis (Hersbach et al., 2020). Sea-level pressure and its difference was chosen in order to provide suitable descriptors of the South Pacific Convergence Zone state and the occurrence of tropical cyclones respectively, that have a major effect on precipitation patterns within this region (Vincent et al., 2011; Mirones et al., 2023). Since the geostrophic approximation is not valid near the equator, we also include the wind components in order to characterize circulation. In order to eliminate redundance and linear dependence among variables to be clustered, we performed a joint Principal Component Analysis (PCA) of all the variables, retaining all PCs explaining up to 80% of total variance (summing up to 45 PCs), prior to clustering. We chose $k$=5 different weather types (WTs) as a trade-off between the representativity of each group (at least 2 years of data in each group) and the minimization of the total within-cluster variance. Additional groups did not lead to significant reductions of within-group variance and resulted in less representative WTs (less than 600 days for the 21-year period 1998-2019), thus potentially affecting the robustness of the conditioned statistical calibration. Furthermore, we assessed the robustness of the obtained classification following a

4-fold cross-validation approach, by partitioning the data in four temporal blocks (1979-1988, 1989-1998, 1999-2008 and 2009-2019). PCA was iteratively trained on each fold and the resulting Empirical Orthogonal Function (EOF) was projected onto the remaining folds. At each iteration, the resulting weather typing yielded consistent results in terms of climatologies and precipitation seasonality. The identification of five distinct daily WTs and their relatively balanced sample sizes ensures a robust conditioning of the calibration process, thereby enhancing the reliability and stability of the calibration results. We provide a summary of the resulting weather type classification in Appendix A. The methodology and main features of the classification are further explained in Mirones et al. (2023).

## 2.3 Bias correction techniques

The calibration techniques used for the adaptive methodology include scaling, empirical quantile mapping (eQM), parametric quantile mapping (pQM), and generalized Pareto distribution quantile mapping (gpQM). A more detailed description of the methods is provided in Appendix B.

The scaling technique is applied to the raw TRMM data by multiplying it with a correction factor. This factor is computed as the ratio between the mean of the predictand (PACRAIN rain gauge measurements) and the mean of the raw TRMM measurements during the training period.

The eQM method is an adaptation of the approach presented in Themeßl et al. (2011), which utilizes empirical cumulative distribution functions (eCDFs) for calibration. In its parametric version (pQM), the QM method relies on the theoretical distribution rather than the empirical one, whose parameters are estimated based on the observed and TRMM data. In particular, here it is assumed that both the observed and simulated intensity distributions can be well approximated by the biparametric gamma distribution (Piani et al., 2010), and therefore both shape and scaling parameters need to be estimated for transfer function fitting.

The gpQM approach also utilizes quantile mapping but incorporates the generalized Pareto distribution (GPD) above a certain threshold (Gutjahr and Heinemann, 2013). The threshold, denoted as $u$, represents the percentile above which the GPD is used to adjust the wet-day distribution. Below the threshold, the distribution is adjusted to a gamma distribution following the pQM method. This method aims to improve the performance of pQM in the upper tail of the distribution, specifically for extreme events. In this work, two different thresholds are selected: the 95th and the 75th percentiles, resulting in the methods named gpQM-95 and gpQM-75, respectively.

## 2.4 Adaptive calibration methodology

### 2.4.1 Calibration model fitting

The adaptive calibration methodology involves the application of scaling, eQM, pQM, gpQM-95, and gpQM-75 (Sec. 2.3) individually to each weather type (WT, Sec. 2.2). Subsequently, the best calibration method for each WT is selected, and the calibrated series are combined to form a unified time series spanning the entire calibration period. For each model fit, the calibrated series are obtained following a cross-validation scheme, aimed at avoiding model overfitting, allowing us to obtain

a more realistic measure of model performance (Efron and Gong, 1983). Under this scheme, the calibration methods are fit and validated $k$ times, considering in turn each of the folds as a test set and training the method with the remaining $k-1$ ones. The resulting $k$-test series are finally joined and validated together in a single series spanning the whole calibration period. In the adaptive calibration approach, the $k$ folds are randomly chosen from each WT separately, $k$ ranging from 2 to 6 folds depending on the number of observations falling within each WT, in order to ensure an approximate sample size of $\sim$300 in each fold in order to ensure a robust fit.

### 2.4.2 Calibration model assessment

In this work, we focus on general validation aspects involving the observed and calibrated-TRMM marginal distributions. In this context, the validation ultimately entails deriving specific precipitation *indices* calculated from both rain gauge observations and calibrated TRMM time series and quantifying the mismatch with the help of suitable *performance measures* (see e.g. Maraun et al., 2015).

There are different types of precipitation indices that can be used to validate a precipitation model, depending on the purpose and scale of the model. The criteria used in this study for choosing precipitation indices for validation are: i) the ability of the index set to capture the relevant aspects of precipitation variability and extremes, such as frequency, intensity and duration, and ii) an index set that is comparable across the different locations and weather types and across different time scales within the study area. A number of precipitation-derived indices are computed to this aim, following the validation framework of the Action Cost VALUE (Gutiérrez et al., 2019, see Table 2), thus enabling a consistent, objective, and shareable evaluation of the quality and performance of different calibration methods (note that here, the term *index* can also refer to a time series, as in the case of correlation as validation measure, that receives as input the raw precipitation times series, Table 2).

However, validation is a multi-faceted process and ranking the different methods using different measures of performance is difficult because there is no single, universal criterion that can capture all aspects of a method's effectiveness. As the different measures emphasize different dimensions of performance (Table 2), such as distributional (e.g. *Skewness* and *Mean*), average precipitation intensity (e.g. Simple Daily Intensity Index -*SDII*-), higher percentile precipitation amount (*P98WetAmount*) or extreme precipitation values for specific return-periods (e.g. *RV20_max*), some measures may be more relevant or important than others, or attain different rankings depending on the method, potentially yielding inconsistent or conflicting outcomes. In order to facilitate a comprehensive evaluation and inter-comparison of each calibration method and WT, we have employed a standardized score calculation methodology able to integrate into one single composite score the different aspects of the validation. Furthermore, specific user-defined weights can be assigned to each validation measure to reflect their relative importance or relevance for the evaluation process, as outlined next.

To determine the best method for each WT, we utilize a Ranking Framework (RF) score, which is based on the methodology described in Kotlarski et al. (2019). The computation of this score involves several steps. Firstly, we calculate the bias of each calibration method with respect to the reference observations by taking the absolute differences between each of the index

values (Table 2) of the reference observations ($X_i$) and the calibrated TRMM series for each method ($Y_{i,j}$):

$$Z_{i,j} = |X_i - Y_{i,j}|. \tag{1}$$

Next, we normalize the bias values obtained from all calibrations ($j$) for a given index ($i$), such that lower values are considered better by the normalization:

$$Z'_{i,j} = 1 - \frac{Z_{i,j} - Z_{i,\min}}{Z_{i,\max} - Z_{i,\min}}. \tag{2}$$

Finally, the RF score for each method is calculated as the average of the normalized values for all indices:

$$RF_j = \frac{1}{N} \sum_{i=1}^{N} w_i \cdot Z'_{i,j} \quad \text{where} \quad \sum_{i=1}^{N} w_i = 1 \tag{3}$$

and $N$ represents the total number of indices evaluated ($N = 10$, see Table 2). Thus, in the calculation of the score, it is possible to incorporate arbitrary (unit normalized) weights $w_i$ for the normalized climate indices. Normalization of weights is a straightforward method where we assign weights based on criteria such as the importance or impact of the validation measure within the validation procedure. For example, we might assign a higher value to performance in extreme indices. Each weight is then divided by the total sum of all weights to ensure they sum to one. This allows for explicit consideration of validation aspects that may carry greater importance, with higher weights assigned to those aspects to determine the final score. The assigned weights have an intuitive interpretation (e.g., a weight of 0.1 for a specific index bias indicates and overall influence of 10% in the final RF score). A common example in many hydrological applications is the significance of accurately representing extreme precipitation events following calibration. Therefore, specific extreme indices (e.g., *P98Wet* or *P98WetAmount*, as shown in Table 2) can be given a higher relative weight in their contribution to the overall score, thereby reflecting their increased relevance in the calibration method ranking process. It is important to note that under this validation framework, the ranking of calibration methods is sensitive to arbitrary decisions in i) the battery of validation indices used and ii) the weight assigned to each measure to compute the overall score. We present a flexible validation framework that can be adapted to various impact applications and research objectives. Nevertheless, end-users can also define their own validation sets according to their specific research questions. With a suitable battery of validation indices, an equally weighted scheme is often a suitable alternative, particularly when aiming to avoid arbitrary choices in index weighting.

All the calibration methods have been run using the implementation available in the package *downscaleR* (Bedia et al., 2020) from the open-source *climate4R* framework for climate data analysis and visualization (Iturbide et al., 2019). The different evaluation indices presented in Table 2 have been computed using the standard definitions of the VALUE Framework (Maraun et al., 2015), which are implemented in the R package *VALUE*[2].

---

[2]https://github.com/SantanderMetGroup/VALUE

**Table 2.** Summary of the validation indices and measures used in the study, along with their corresponding codes as defined in the VALUE reference list (`http://www.value-cost.eu/validationportal/app/#!indices`). The indices and measures serve as evaluation metrics for assessing the performance and accuracy of the calibration techniques in the study.

| Code | Description | Type |
|------|-------------|------|
| Skewness | Skewness | index |
| Mean | Mean | index |
| SDII | Mean wet-day ($\geq$ 1mm) precipitation | index |
| R10 | Relative frequency of days with precip $\geq$ 10mm | index |
| R10p | Precipitation amount falling in days with precip $\geq$ 10mm | index |
| R20 | As R10, but considering a 20mm threshold | index |
| R20p | As R10p, but considering a 20 mm threshold | index |
| P98Wet | 98th percentile of wet ($\geq$1 mm) days | index |
| P98WetAmount | Total amount above 98th percentile of wet ($\geq$1 mm) days | index |
| RV20_max | Maximum Daily precipitation for a 20-year Return Value | index |
| absolute bias | | measure |
| relative bias | | measure |
| Spearman's rank correlation | | measure |

## 3 Results and discussion

### 3.1 Standard and weather-type conditioned calibration method intercomparison

To obtain a comprehensive evaluation of the method's overall performance, we initially focus on an unconditioned intercomparison, referred to as "standard calibration" hereafter. This evaluation involves assessing the method across the entire time series without considering different weather types. The preliminary findings indicate the presence of low to moderate (negative) biases in the TRMM product. As an illustrative example, we present the results obtained at the Kolopelu station in Fig. 1 (the upper triangle represents the standard calibration results), which are representative of the overall outcomes observed at other locations (the corresponding plots for the remaining rain gauge locations are included in Appendix B). While certain TRMM indices show negligible biases compared to the rain gauge stations (such as *Skewness* and *SDII*), others exhibit significant relative biases, particularly for representing high precipitation events (such as *R10p*, *R20p*, or *P98_WetAmount*). These findings highlight the necessity of applying some form of calibration to enhance the accuracy of TRMM for impact studies. In the same vein, we include the ERA5 precipitation series to provide a benchmark to contextualize the biases of satellite data. This comparison helps to highlight the strengths of the satellite products while also underscoring the importance of calibration against a reliable reference, as satellite product biases remain an issue.

Under the assumption that the satellite and radar error may be proportional to the magnitude of the rain rate (Aghakouchak et al., 2009), scaling has been previously used for TRMM correction (see e.g. Islam et al., 2010; Almazroui, 2011). However, it was found to be ineffective in mitigating biases in most specific indices related to intense precipitation events and some others related to mean precipitation such as *SDII*. Instead of yielding and improvement, scaling had an overall deleterious effect,

underscoring the critical importance of considering alternative calibration techniques better suited to TRMM adjustment. As shown in Fig. 1, the scaling step introduces additional bias to the raw TRMM data, specially for *SDII*, *R10p R20p* and *P98Wet*.

The scaling step also changes the sign of the bias in these cases, but consistently increases its absolute value, both in the standard (not conditioned) and the WT-conditioned approaches.

As Figure 1 shows, overall the quantile-mapping based techniques attain similar performance, although this may vary by index or technique. The relative bias of the indices is also similar for both standard and conditioned calibration. This supports Mirones et al. (2023)'s finding that EQM outperforms scaling, with minor differences between unconditioned and WT-

250 conditioned calibration in bias. This also applies to the parametric quantile mapping variants pQM and gpQM, for different exceedance thresholds, which are additional methods introduced in this study.

The final column in Figure 1 compares the best-performing conditioned technique in each case with the adaptive methodology, combining the optimal calibration technique for each weather type individually. In general, the adaptive methodology surpasses the results achieved by the best-conditioned calibration. The most notable reduction in relative bias is observed

in indices that measure high rainfall amounts, such as *R10p*, *R20p*, or *P98WetAmount*. This improvement is significant because conditioned calibration alone did not exhibit substantial enhancements, except for specific cases involving techniques like gpQM. However, only three indices (*Mean*, *SDII*, and *P98Wet*) did not show improvement with the adaptive approach and remained nearly unchanged. In summary, the overall results indicate that the adaptive calibration method offers improved adjustment in the upper tail of the distribution, which is where TRMM exhibits the most significant biases. This calibration

methodology facilitates enhanced adjustment for extreme precipitation events, with a specific focus on high precipitation indices. Next, in Sec. 3.2, we present a more in-depth analysis of the detailed results obtained from the adaptive calibration approach.

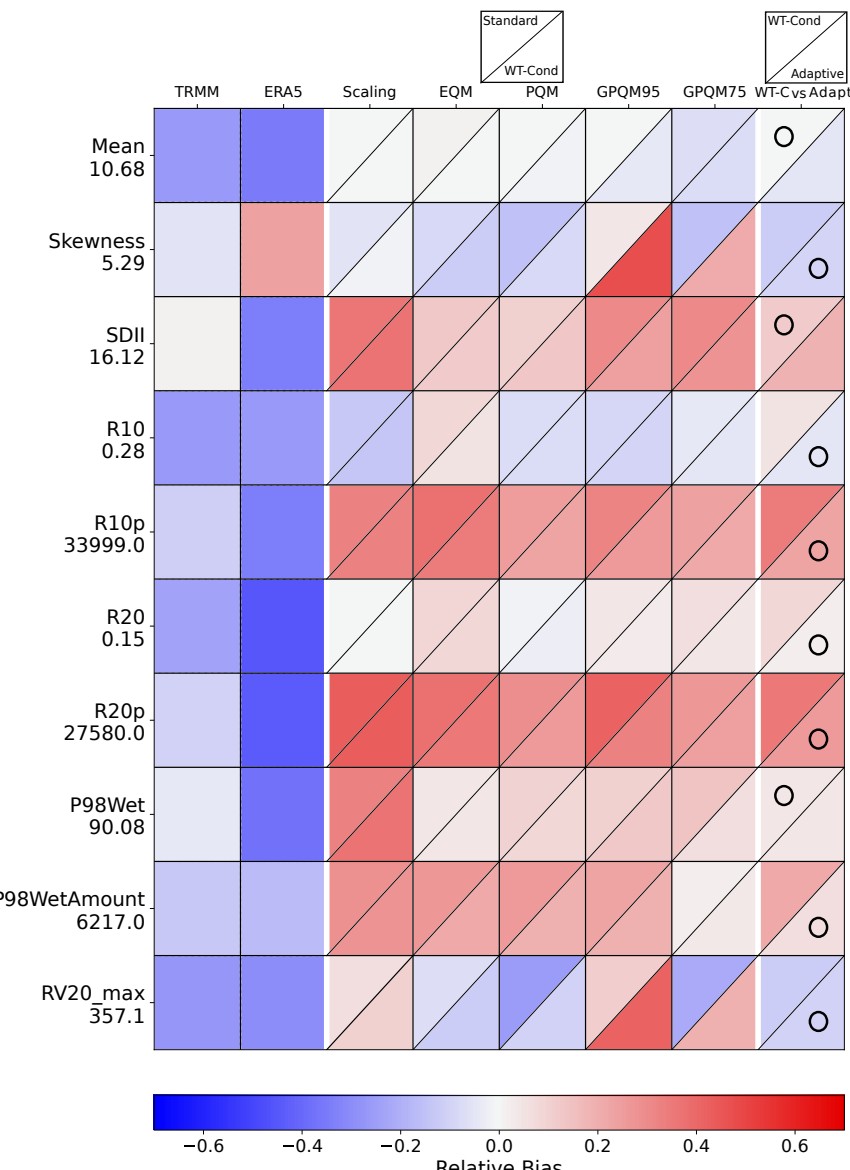

**Figure 1.** Relative Biases of the climate indices used for validation (Table 2 of raw TRMM data and TRMM-calibrated data, at the Kolopelu Station grid box (Table 1). As an additional reference, we include in the second column the biases of the ERA5 reanalysis raw precipitation data (Hersbach et al., 2020). The calibration techniques are scaling, eQM, pQM, GPQM-95, and GPQM-75 (Sec. 2.3). For each method plot cell, the upper triangle displays the relative bias of the standard calibration, while the lower triangle represents the WT-conditioned approach. The last column shows a comparison between the relative bias of the best performing WT-conditioned technique (eQM at the Kolopelu site) *vs.* the 'adaptive' calibration. The circle indicates the best-performing approach with the lowest relative bias. The Y-axis labels show the actual index values from the rain gauge observations beneath the index names.

## 3.2 Adaptive calibration

We introduce the RF score as a comprehensive measure that accounts for the various indices described earlier (Table 2) to evaluate the performance of the adaptive approach in comparison to the standard WT-conditioned method. This measure allows for an easier ranking of methods. Figure 2 presents the summarized results, considering both unweighted and weighted RF values, where 'unweighted' here refers to a version in which all evaluation metrics are equally weighted, and thus no specific weights are assigned by the user. The latter emphasizes the significance of extreme indicators in the evaluation process, as discussed in Section 2.4.

First of all, we undertake a method intercomparison using the unweighted RF score as a ranking measure (Fig. 2a). In most stations the adaptive calibration outperforms the standard and WT-conditioned techniques, with some exceptions (Port Vila and Nuu'uli), in which the adaptive approach has a similar performance than the simpler ones. On the contrary, the adaptive method outperforms the standard and WT-conditioned calibration at other locations, such as Kolopelu, Alofi, or Raoul Island. Here, the adaptive calibration yields much higher RF scores than the best scores achieved by the other methods. While the score for the best standard technique at these stations exceeds 0.60, the adaptive calibration achieves values between 0.80 and 0.90, representing an improvement of 33-50%. This significant enhancement in calibration, demonstrates an overall improvement that justifies the application of the adaptive method. Furthermore, it is noteworthy that in the worst-case scenario, the adaptive approach will nearly match (and never significantly impair) the performance of the calibration. Furthermore, the limited spatial variability across locations observed in the adaptive method (Fig. 2, boxplots) underpins its potential for a robust application across different locations. This characteristic holds importance in hydrological studies where spatial consistency between locations at the basin level is typically desired.

As mentioned earlier, it is also possible to assign arbitrary weights to the indices involved in the RF score, giving more importance to specific precipitation characteristics, such as the representation of extremes. In this study, we selected index weights that prioritize extreme precipitation indices (Table B2). This weighting aims to guide the calibration towards better adjustment in the upper tail of the distribution, thereby achieving improved correction for extreme precipitation events beyond a certain threshold. In this way, the increased influence of extreme indices favors methods like gpQM, which specialize in adjusting the upper tail using a GPD (Generalized Pareto Distribution). The findings are illustrated through the boxplots presented in Figure 2, showing that the scores of gpQM95 and gpQM75 exhibit higher values in the weighted version (Figure 2b) compared to the unweighted version (Figure 2a). Moreover, the differences in their performance can be visualized in Fig. 3.

The analysis of the RF weighting configuration demonstrates its dual impact: not only does it affect the overall score, but it also influences the selection of techniques for each weather type (Fig. 3). In our weighting scheme, which prioritizes performance in extreme precipitation indices, the gpQM approaches often appear as the best choice, for instance at Aoloau site for WTs 1 to 3, Kolopelu (WTs 1 and 2) or Alofi (WT1, Fig. 3). This outcome indicates the suitability of the gpQM technique over the conventional eQM and pQM methods when the representation of extreme indices is prioritized. For instance, at Aoloau the overall adaptive calibration RF score improves from approximately 0.65 to 0.85, representing a 30% increase, with the best standard calibration method changing from eQM or pQM to gpQM75 in WT1, WT2 and WT3 (Fig. 3).

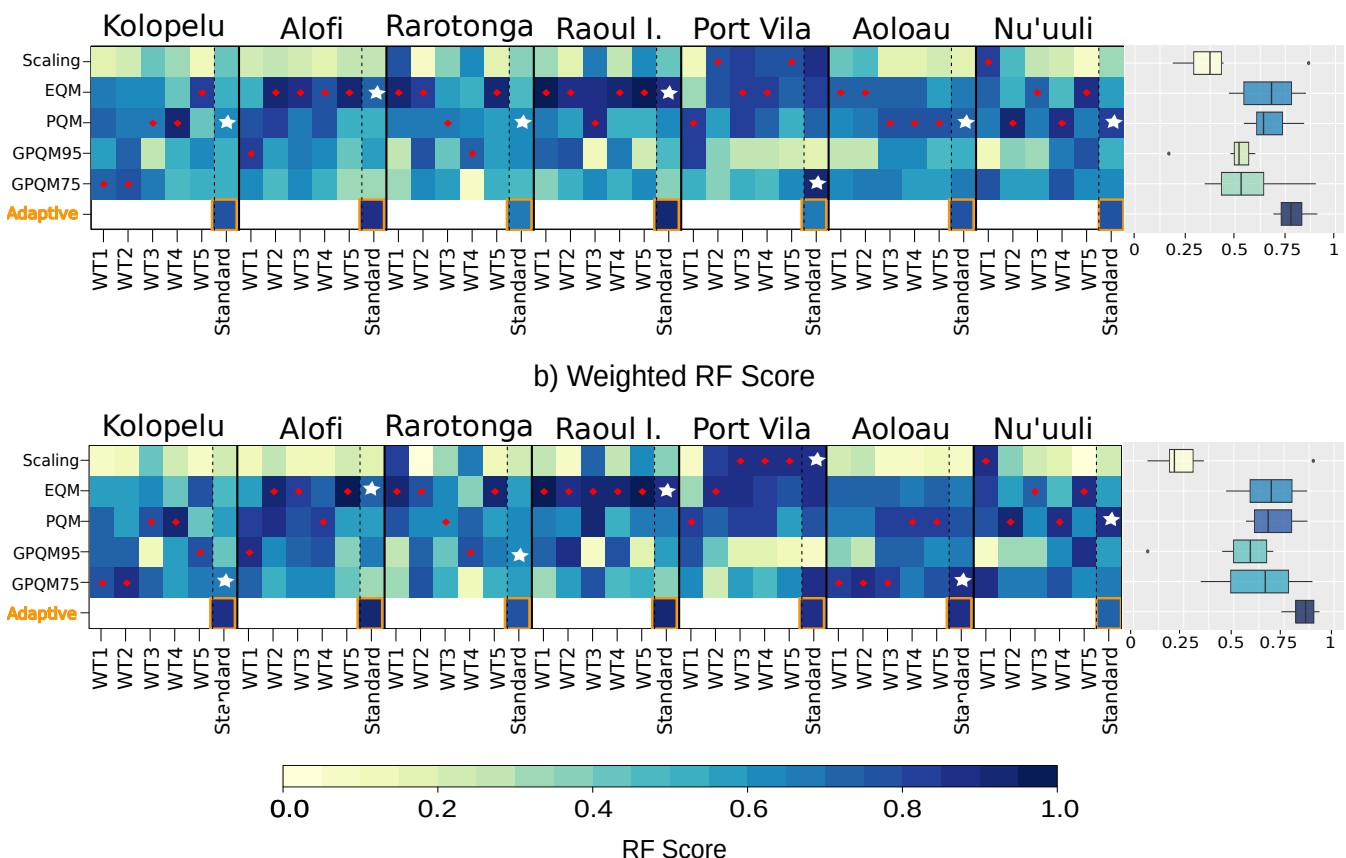

**Figure 2. (a)**: Ranking Framework (RF) score results of the adaptive calibration for each WT and site. The red dots indicate the best method for the corresponding WT, while the white stars represent the best method for the standard calibration (the same technique applied over the entire period without conditioning). On the right side, the standard approach score for each method is represented in the box plot (their mean represented by the color of the boxes). **(b)**: Similar to **(a)**, but with the addition of different weights (refer to B2) in the computation of climate indices for the RF score.

In the same vein, at the Port Vila station in Fig. 2a, the adaptive calibration score is lower compared to the score of gpQM75. However, when applying weighting with a focus on high rainfall indices (Figure 2b), the adaptive calibration undergoes enhancements and achieves a competitive score higher than the unweighted version. It is worth noting that while the inclusion of weights leads to changes and improvements in the adaptive calibration for certain stations, it has no effect on others. For instance, stations like Rarotonga or Nu'uuli exhibit no changes in the composition of the adaptive calibration, regardless of the weighting applied. The particular case of Rarotonga site emphasizes the fact that model assessment is sensitive to user choices, such as the set of indices and measures used for validation and, as in this case, the weights assigned to each of them. As a

result, when more weight is assigned to the performance of extreme precipitation indices (weighted RF evaluation), the adaptive calibration approach performs better than the standard calibration with the best-performing method (PQM, RF~0.75). On the contrary, the standard unconditioned calibration using the PQM method performs (marginally) better under the unweighted RF scheme. In the case of Nu'uli station, the adaptive calibration did not improve the performance, although it also attained a similar high RF score (~0.85/0.75 unweighted/weighted RF scores, Fig. 2a/b). Compared to other locations in Figs. B1-B6 (Appendix C) and Fig. 1 (Kolopelu), Nu'uli has moderate raw TRMM biases that are effectively corrected by both empirical and parametric quantile mapping Fig. B6. gpQM performs similarly in general, but fails in reproducing distributional skewness and *RV20_max* in the case of the adaptive approach.

Therefore, the results show that adaptive method attains an overall better performance in all stations, as highlighted in the boxplots in Figure 2. Additionally, the adaptive calibration demonstrates a narrower interquartile range (IQR) compared to the other methods in Figure 2a, indicating lower variability. Only gpQM95 calibration shows a narrower IQR range, but obtaining significantly lower scores. We attribute this result to the limited robustness of the fit of the extreme function due to the high percentile threshold, which greatly reduces the sample size (see Table B1). The other methods individually considered exhibit wider RF variability ranges and lower values as compared to the adaptive approach.

Quantile mapping can preserve the relative changes in the simulated data, such as trends and patterns (see e.g. Casanueva et al., 2018), while scaling may distort them by applying a constant factor. As a result, scaling shows limitation in representing some of the extreme precipitation indices such as *p98Wet* and *RV20_max* in most of the locations, as well as the *SDII* (see Figs. B1-B6 and Fig. 1 -Kolopelu-). On the other hand eQM and pQM show an overall good performance in most precipitation indices, and eQM is most often the best method for the "standard" calibration (Fig. 2).

The gpQM method has the potential to improve the modeling of local extreme precipitation by better adjusting extreme events above a given percentile (75 and 95 in this study), by fitting a generalized Pareto Distribution on the threshold exceedances as the transfer function for these points (see e.g. Vrac and Naveau, 2007). This may happen, for instance, in Weather Types 1 and 2, which are associated with enhanced tropical cyclone frequency and extreme precipitation events (Mirones et al. (2023), and Fig. A), and it is shown at Kolopelu (Fig. 2a/b for unweighted/weighted RF score) and Aoloau (Fig. 2b), where gpQM75 is the best-performing calibration method, or WT1 at Alofi (with gpQM95). The limitation, however, arises from the sample size required to perform a robust fit; this is a major challenge in the adaptive approach, since only the observations of the given WT are used, and it is exacerbated for higher percentile thresholds that further limit the number of points for a robust fit. This limitation also exists for the other quantile mapping techniques, although, in this case, the number of observations, even for the less frequent weather type (approximately 300 days for the calibration period in this study, see Sec. 2.4.1), can ensure a robust fit. Nevertheless, scaling requires less data and is easier to implement than quantile mapping. The assumption of scaling of a linear relationship between raw satellite and observed precipitation, even though may not hold true in most occasions, can still be reasonable (Aghakouchak et al., 2009) yielding a good approximation in some occasions (see e.g. the overall RF results obtained in Port Vila in Fig. 2).

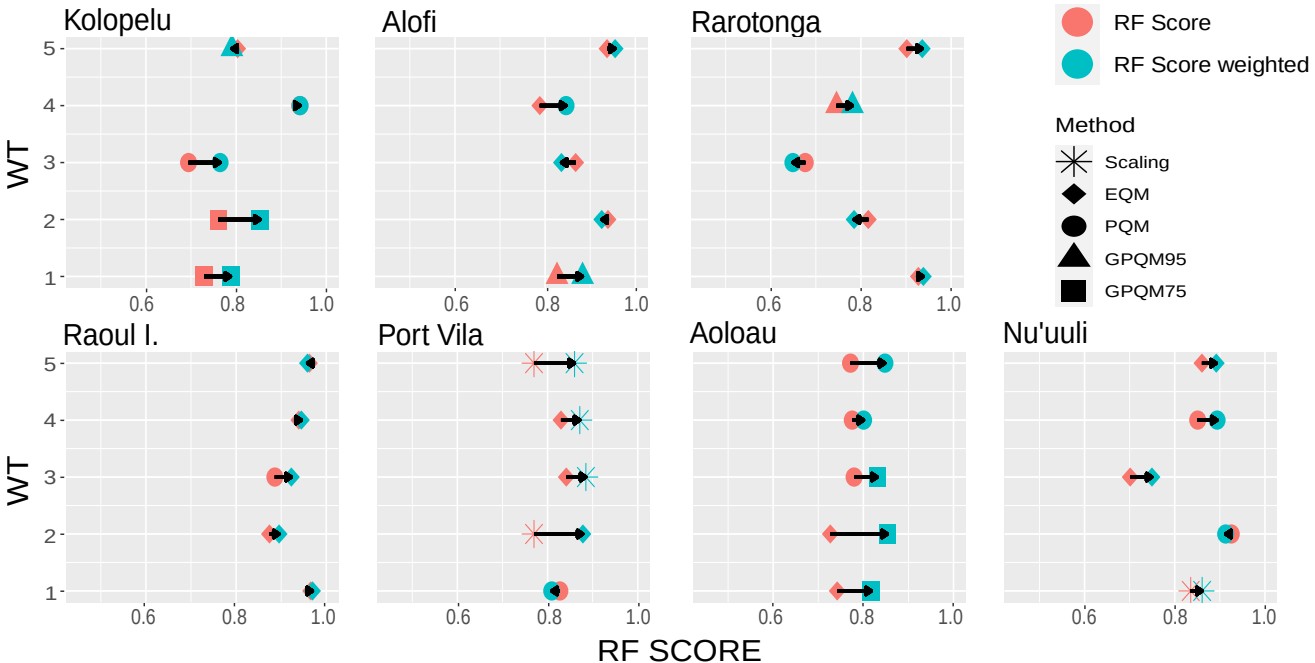

**Figure 3.** Differences between unweighted and weighted RF scores attained for each weather type at each of the target locations. The technique associated to the highest score is also indicated by the key of symbols.

## 4 Conclusions

We intercompared a range of bias-adjustment techniques for the calibration of daily Tropical Rainfall Measuring Mission (TRMM) precipitation data, building upon a set of rain-gauge stations scattered across the South Pacific region, spanning the
340 period 1998–2019. The calibration techniques evaluated in this study include empirical quantile mapping (eQM), parametric quantile mapping (pQM), generalized Pareto distribution quantile mapping (gpQM) and scaling, the latter used as a benchmark since it is the most common and simple approach for this task. An adaptive calibration methodology has been developed based on weather type (WT) conditioning, which selects the best-performing calibration technique for each specific WT according to a user-defined weighted set of precipitation indices for validation.

The adaptive calibration method improves upon the results obtained with the standard WT-conditioned methodology presented in Mirones et al. (2023) or at least, in the worst case, it maintains the calibration performance. Notably, the most substantial improvements are observed in accumulated precipitation indices, specifically *R10p*, *R20p* and *P98WetAmount*. These indices hold significance in hydrological modeling and climate impact studies, making the attained improvements in the calibration particularly relevant. Thus, the method showcases competitive performance in effectively calibrating the TRMM data
at the target stations, thereby promoting consistency in the results across diverse locations. This adaptability further enhances the overall calibration accuracy, by adjusting its bias according to the specific precipitation regimes prevalent in each weather type. Furthermore, the capability to customize the calibration by applying arbitrary weights to specific indices offers increased

flexibility in determining the optimal combination of methods that align with the unique characteristics of each site and research objectives. In conclusion, this new adaptive calibration methodology refines and improves the accuracy of precipitation data from indirect sources, such as the TRMM database or similar products. This enhances the reliability of hydrological and climate impact assessments based on these data. In the supplementary material, we provide the data and a reproducible documented notebook, to aid in the application of the method.

*Code and data availability.* An interactive notebook associated with this study is available at the following link: https://github.com/SantanderMetGroup/notebooks/tree/2023_TRMM_adaptiveCal/2023_adaptiveCalibration. This notebook provides a comprehensive illustration of the entire adaptive calibration process, including available data download from an open repository and the computation of both standard and adaptive calibration RF scores.

*Acknowledgements.* We thank two anonymous referees for insightful comments in previous versions of the manuscript. This paper is part of the R+D+i project CORDyS (PID2020-116595RB-I00) with funding from the Spanish Ministry of Science MCIN/AEI/10.13039/501100011033. O.M. has received research support from grant PRE2021-100292 funded by MCIN/AEI/10.13039/501100011033. We also thank our colleague Dr. Javier Díez-Sierra for sharing code snippets used to develop some of the figures. We are grateful to our colleagues F.J. Méndez, S. O. van Vloten, A. Pozo and L. Cagigal for fruitful discussions on weather typing applications in the South Pacific region.

*Competing interests.* The authors declare no competing interests.

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

## Appendix A: Weather typing results

Next, we provide a visual summary of the weather typing classification undertaken using ERA5 reanalysis (Sec. 2.2). Here, we include a summary figure of the resulting classification (Fig. A1) in order to include additional relevant information for the analysis presented in this study, such as the occurrence of tropical cyclones (TCs), as relevant features significantly contributing

to total annual precipitation and directly related with extreme precipitation events in the region. The TC track record for the study area has been obtained from the International Best Track Archive for Climate Stewardship (IBTrACS - v4.0) database (Knapp et al., 2010)[3].

While the seasonal pattern of the classification is depicted, interannual variability fluctuations also emerge in this visualization. A few years stand-out as particularly cyclonic (e.g. 1981, 1997, 1998) being most of these TC events associated to

weather types 1 and 4 (e.g. years 1983, 2003 and 2010) and mostly restricted to the period December-April (Fig. A1b). The exceptional year 1997 extends the TC activity to May-July and starts earlier, in October, while the second strongest (1981) does not extend the season but exhibits an increased TC activity in the period December-March (Fig. A1a). Notably, both years are characterized by experiencing the two strongest El-Niño events of recent decades[4], together with 2015-2016, that also exhibits a remarkable TC frequency (Fig. A1b). The interested reader is referred to the weather typing methodology and main results

presented in more detail in Mirones et al. (2023).

## Appendix B: Bias correction methods formulas and complementary information

Here we provide a detailed description of the correction methods used in the study. These methods aim to improve the accuracy of the TRMM rainfall data by incorporating information from PACRAIN rain gauge measurements, used as predictand. The calibration techniques are next described:

---

[3] All data are open access through a dedicated URL (https://www.ncdc.noaa.gov/ibtracs/index.php?name=ib-v4-access)

[4] El Nino and La Nina Years: NOAA Physical Sciences Laboratory [WWW Document], n.d. URL https://psl.noaa.gov/enso/climaterisks/years/top24enso.html (accessed 2.12.24).

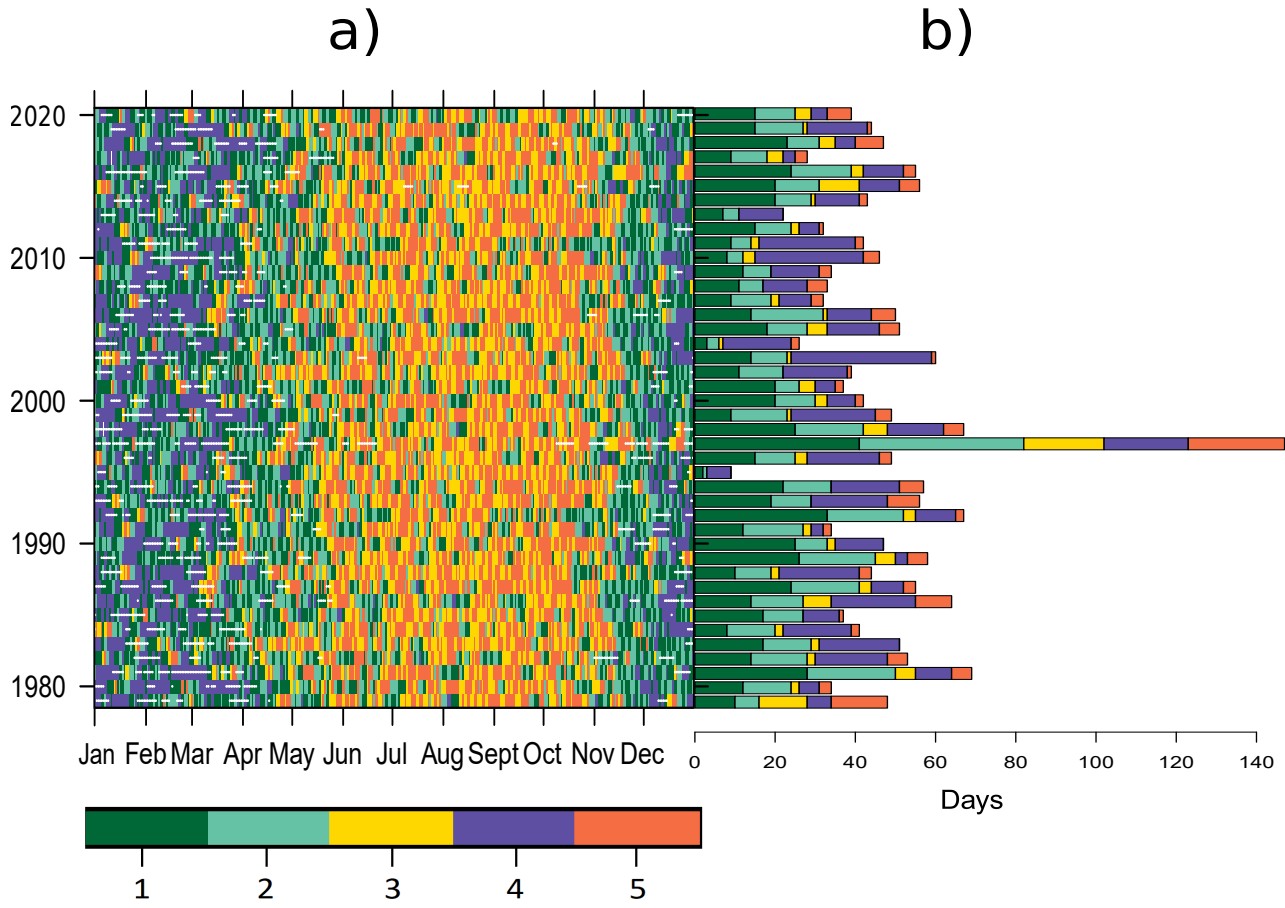

**Figure A1.** (a): Daily time series classification of weather types for the extended 41-year period 1979-2020. TCs occurrences are represented by the white dots. (b): Absolute number of days affected by TCs by year and WT.

Equation B1 presents the scaling method, where $\hat{p}_{trmm}$ represents the corrected TRMM rainfall, $p_{rg}$ and $p_{trmm}$ denote the PACRAIN rain gauge and raw TRMM measurements, respectively, and $\bar{P}_{rg}$ and $\bar{P}_{trmm}$ are the means of the $p_{rg}$ and $p_{trmm}$ series.

$$\hat{p}_{trmm} = p_{trmm} \frac{\bar{P}_{rg}}{\bar{P}_{trmm}} \tag{B1}$$

Equation B2 describes the empirical quantile mapping (eQM) method. Here, $\hat{X}_{t,i}$ represents the corrected value for a specific
day and grid, $\hat{F}^{trmm}_{doy,i}$ and $\hat{F}^{rg}_{doy,i}$ are the empirical cumulative distribution functions (eCDFs) for TRMM and PACRAIN, respectively, corresponding to the given day of the year ($doy$), and $X_{t,i}$ is the uncorrected value.

$$\hat{X}_{t,i} = \hat{F}^{rg^{-1}}_{doy,i}(\hat{F}^{trmm}_{doy,i}(X_{t,i})), \tag{B2}$$

Equation B3 presents the parametric quantile mapping (pQM) method. $F_{doy,i}^{trmm}$ and $F_{doy,i}^{rg}$ represent the assumed theoretical distributions for TRMM and PACRAIN, respectively, and $X_{t,i}$ is the uncorrected value.

$$\hat{X}_{t,i} = F_{doy,i}^{rg^{-1}}(F_{doy,i}^{trmm}(X_{t,i})), \tag{B3}$$

Lastly, Equation B4 outlines the generalized parametric quantile mapping (gpQM) method. It uses a combination of gamma and generalized Pareto distribution (GPD) to correct the TRMM rainfall values based on their percentiles. $F_{doy,i}^{trmm,gamma}$ and $F_{doy,i}^{rg,gamma}$ are the gamma cumulative distributions for TRMM and PACRAIN, while $F_{doy,i}^{trmm,GPD}$ and $F_{doy,i}^{rg,GPD}$ represent the GPDs for TRMM and PACRAIN, respectively. The threshold of the 95th percentile is used to differentiate between the two distributions.

$$\hat{X}_{t,i} = \begin{cases} F_{doy,i}^{rg,gamma^{-1}}(F_{doy,i}^{trmm,gamma}(X_{t,i})) & \text{if } X_{t,i} < \text{95th percentile} \\ F_{doy,i}^{rg,GPD^{-1}}(F_{doy,i}^{trmm,GPD}(X_{t,i})) & \text{if } X_{t,i} \geq \text{95th percentile} \end{cases} \tag{B4}$$

**Table B1.** Overview of observations (days) for each WT across multiple stations. Each row corresponds to a specific station, while the columns represent different WTs. The table displays the total number of observations recorded for each WT, along with the corresponding 75th and 95th percentiles.

| Station | WT1 | | | WT2 | | | WT3 | | | WT4 | | | WT5 | | |
|---|---|---|---|---|---|---|---|---|---|---|---|---|---|---|---|
| | N | P75 | P95 | N | P75 | P95 | N | P75 | P95 | N | P75 | P95 | N | P75 | P95 |
| Kolopelu | 966 | 242 | 48 | 775 | 194 | 39 | 851 | 213 | 43 | 826 | 206 | 41 | 306 | 76 | 15 |
| Alofi | 1140 | 285 | 57 | 932 | 233 | 47 | 1023 | 256 | 51 | 917 | 229 | 46 | 437 | 109 | 22 |
| Rarotonga | 1105 | 276 | 55 | 916 | 229 | 46 | 1055 | 264 | 53 | 957 | 239 | 48 | 391 | 98 | 20 |
| Raoul Island | 1327 | 332 | 66 | 1058 | 264 | 53 | 1158 | 290 | 58 | 1076 | 269 | 54 | 458 | 114 | 23 |
| Port Vila | 1084 | 271 | 54 | 884 | 221 | 44 | 1010 | 252 | 50 | 849 | 212 | 42 | 370 | 92 | 18 |
| Aoloau | 1091 | 273 | 55 | 874 | 218 | 44 | 999 | 250 | 50 | 826 | 206 | 41 | 428 | 107 | 21 |
| Nu'uuli | 2076 | 519 | 104 | 1721 | 430 | 86 | 1846 | 462 | 92 | 1570 | 392 | 78 | 820 | 205 | 41 |

## Appendix C:  Comparison of IMERG and TRMM dataset biases

In this Appendix we indicate the biases of the IMERG product (together with TRMM and ERA5 reanalysis as reference), as compared against the rain gauge observations at the seven locations of the study. Our analysis shows that the biases observed in TRMM are still present in the newer IMERG products at the selected localities. In Fig. C1, we depict the biases of both TRMM and IMERG using a quantile-quantile plot against the rain gauge records. In Fig. C2, these biases are more accurately quantified in terms of the different validation indices used in this study. The new IMERG products show a moderate overall improvement over the old TRMM product in some sites (Alofi, Raoul Island); however, biases are still present, and in some locations are even higher that TRMM (Port Vila) necessitating calibration against a 'ground-truth' reference.

**Table B2.** Index weights $w_i$ (see Eq. 3) for the weighted RF score calculation in the adaptive calibration technique selection.

| Code | Weight |
| --- | --- |
| Skewness | 0.05 |
| Mean | 0.05 |
| SDII | 0.05 |
| R10 | 0.05 |
| R10p | 0.05 |
| R20 | 0.15 |
| R20p | 0.15 |
| P98Wet | 0.15 |
| P98WetAmount | 0.2 |
| RV20_max | 0.1 |

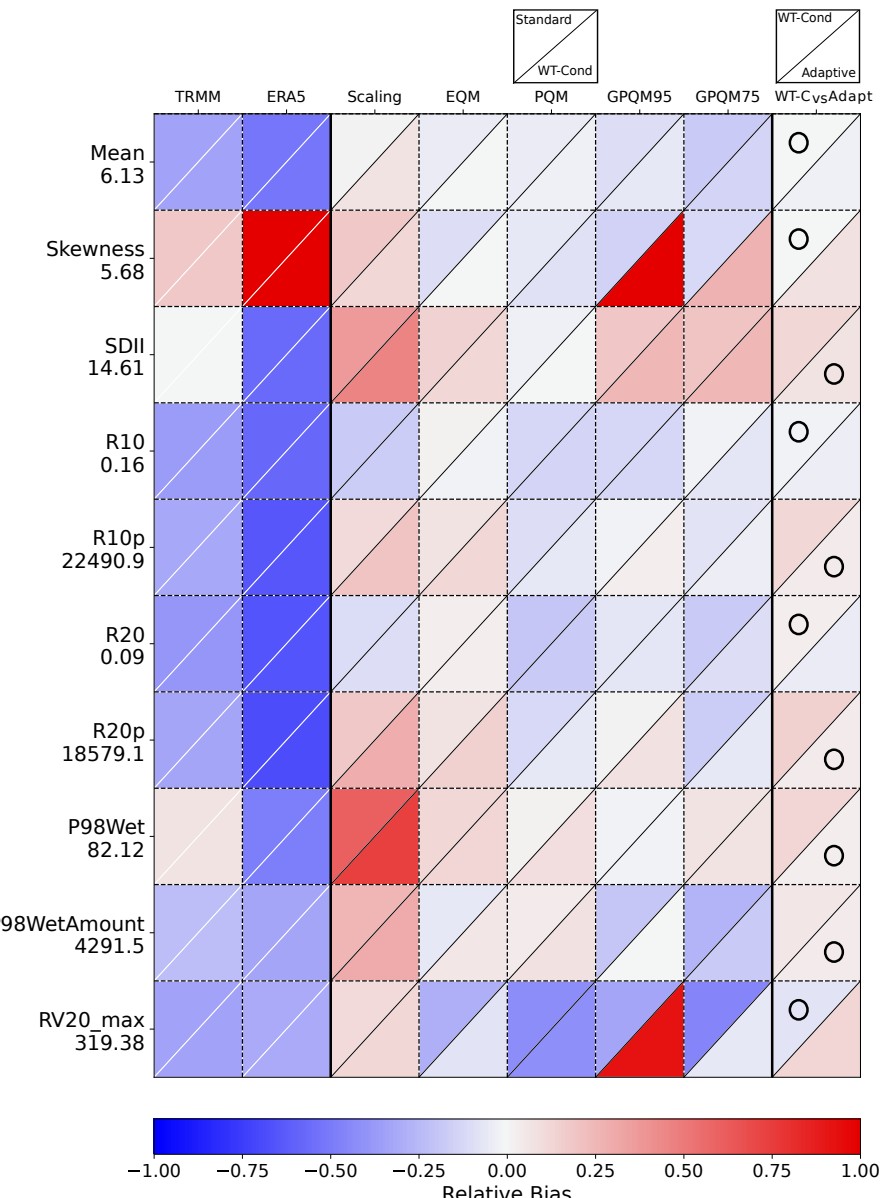

**Figure B1.** Relative Biases of the climate indices used for validation (Table 2 of raw TRMM data and TRMM-calibrated data, at the Alofi Station grid box (Table 1). As an additional reference, we include in the second column the biases of the ERA5 reanalysis raw precipitation data (Hersbach et al., 2020). The calibration techniques are scaling, eQM, pQM, GPQM-95, and GPQM-75 (Sec. 2.3). For each method plot cell, the upper triangle displays the relative bias of the standard calibration, while the lower triangle represents the WT-conditioned approach. The last column presents shows a comparison between the relative bias of the best WT-conditioned technique vs. the adaptive calibration. The circle indicates the best-performing approach with the lowest relative bias. The Y-axis labels show the actual index values from the rain gauge observations beneath the index names.

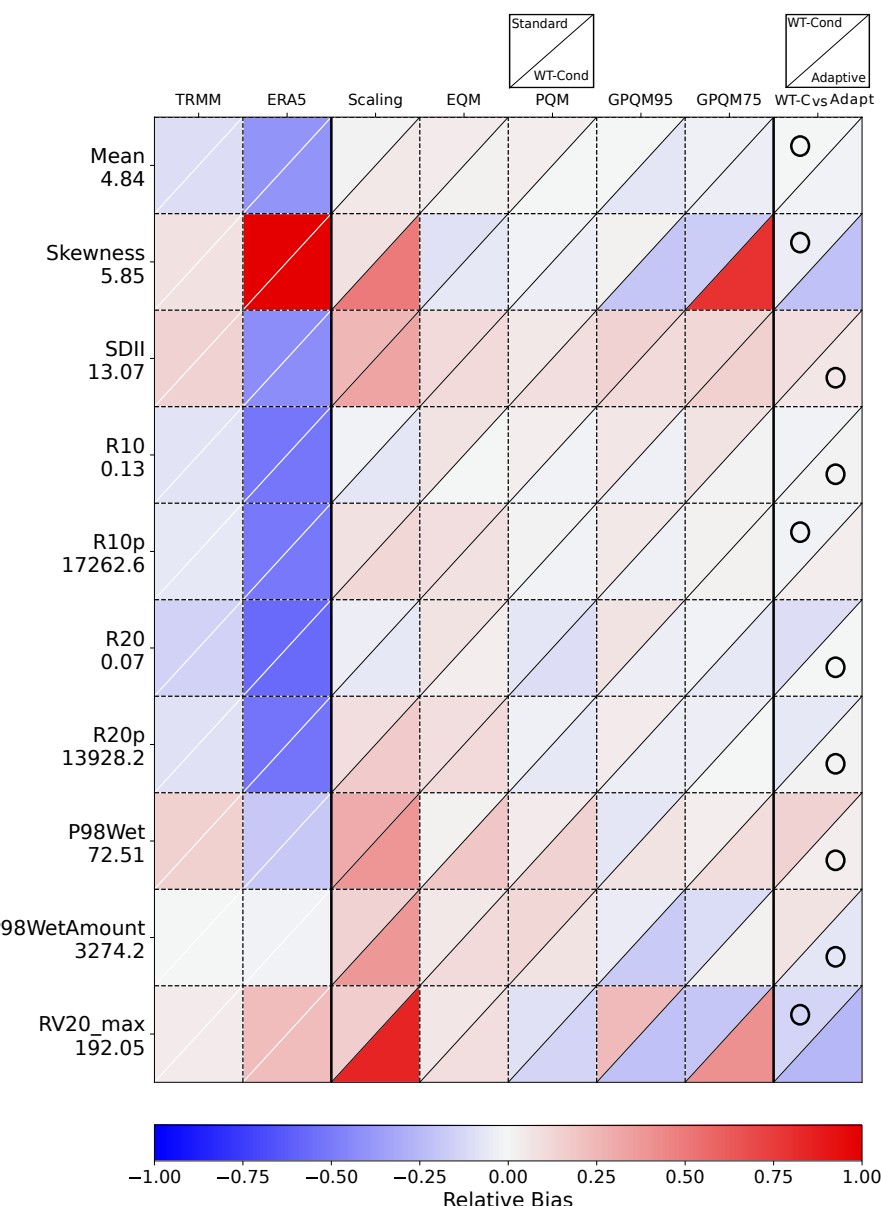

**Figure B2.** Same as Fig. B1, but for the Rarotonga rain gauge location.

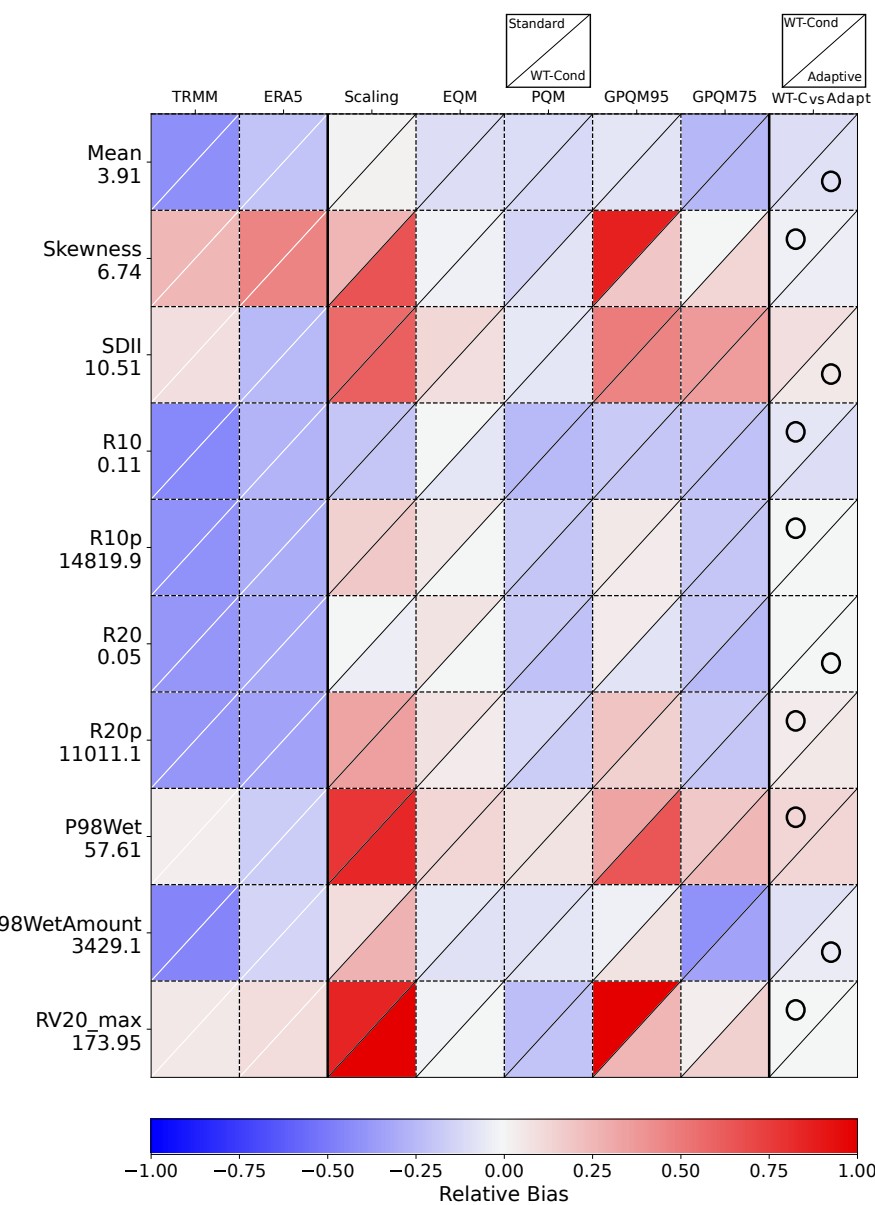

**Figure B3.** Same as Fig. B1, but for the Raoul Island rain gauge location.

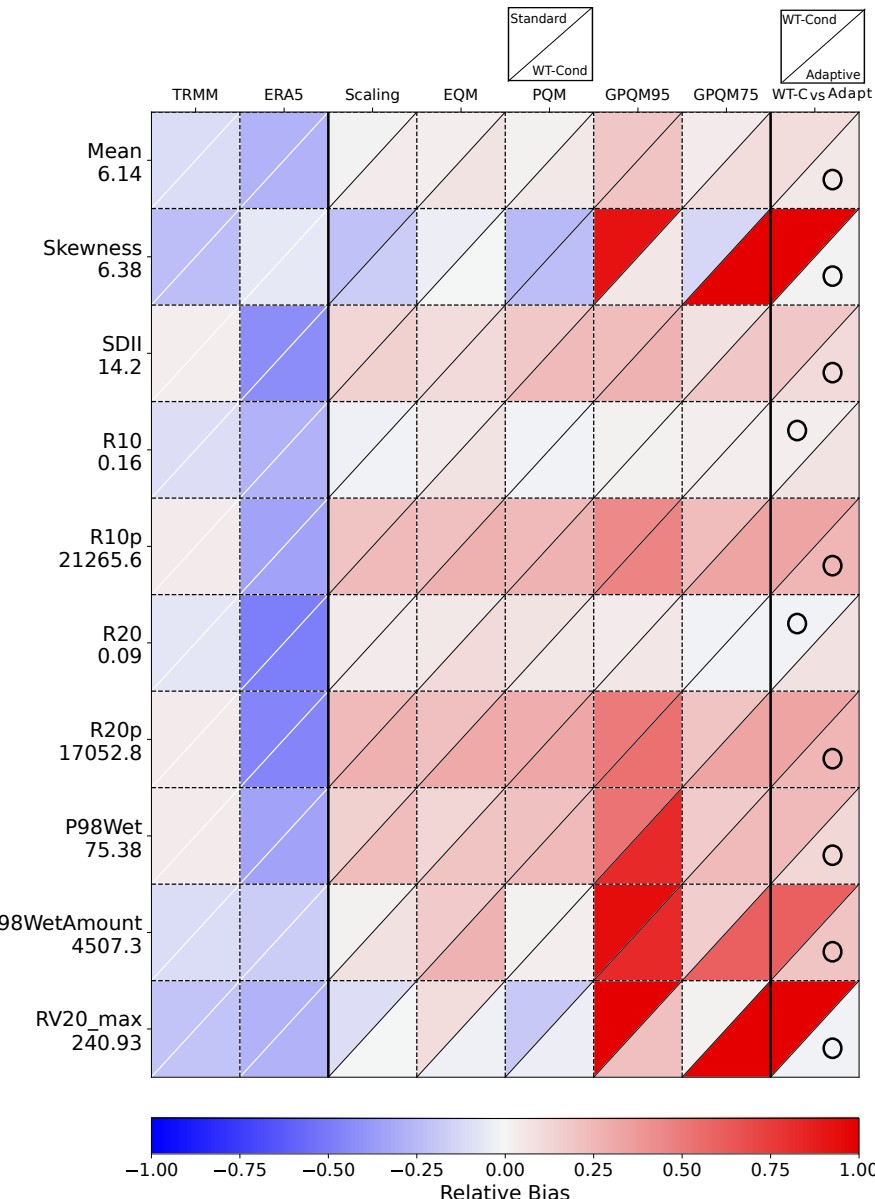

**Figure B4.** Same as Fig. B1, but for the Port Vila rain gauge location.

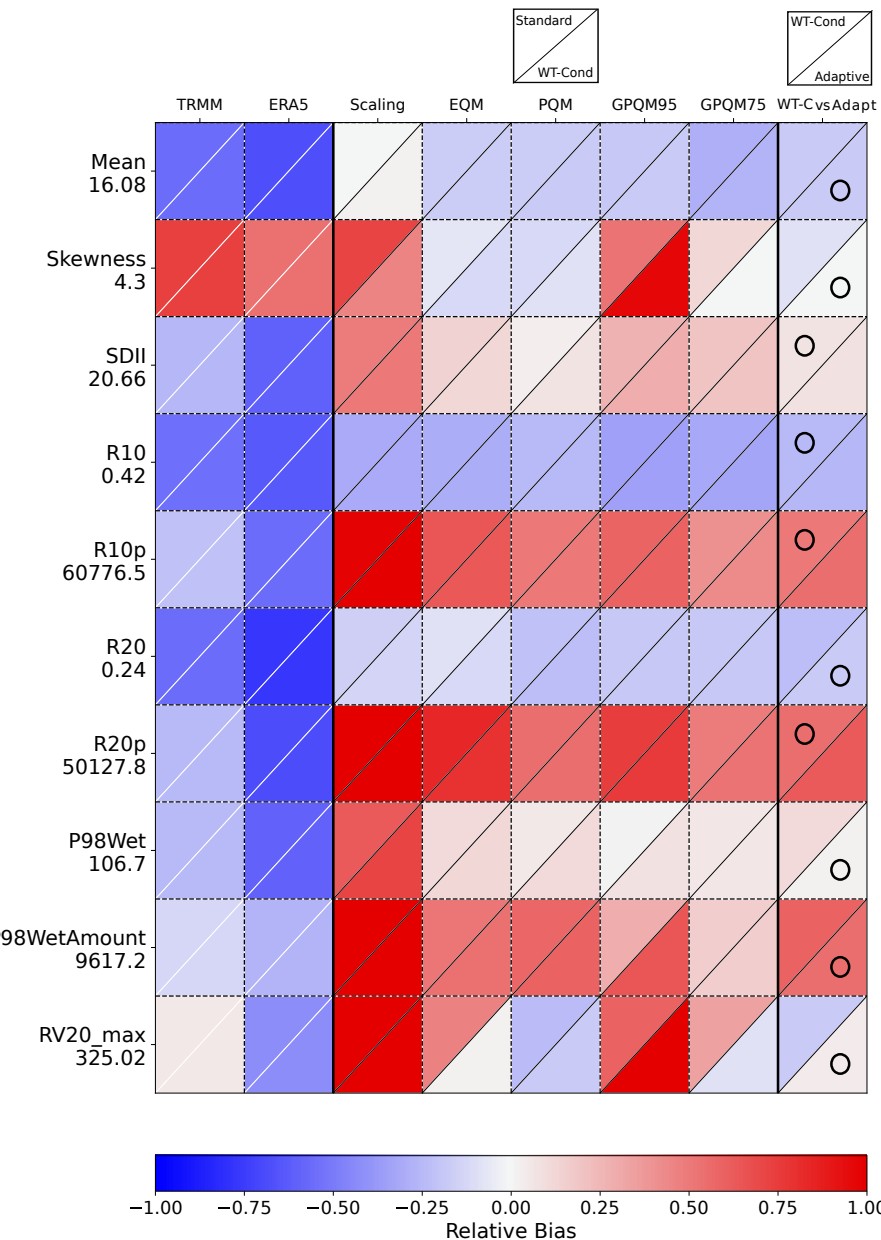

**Figure B5.** Same as Fig. B1, but for the Aoloau rain gauge location.

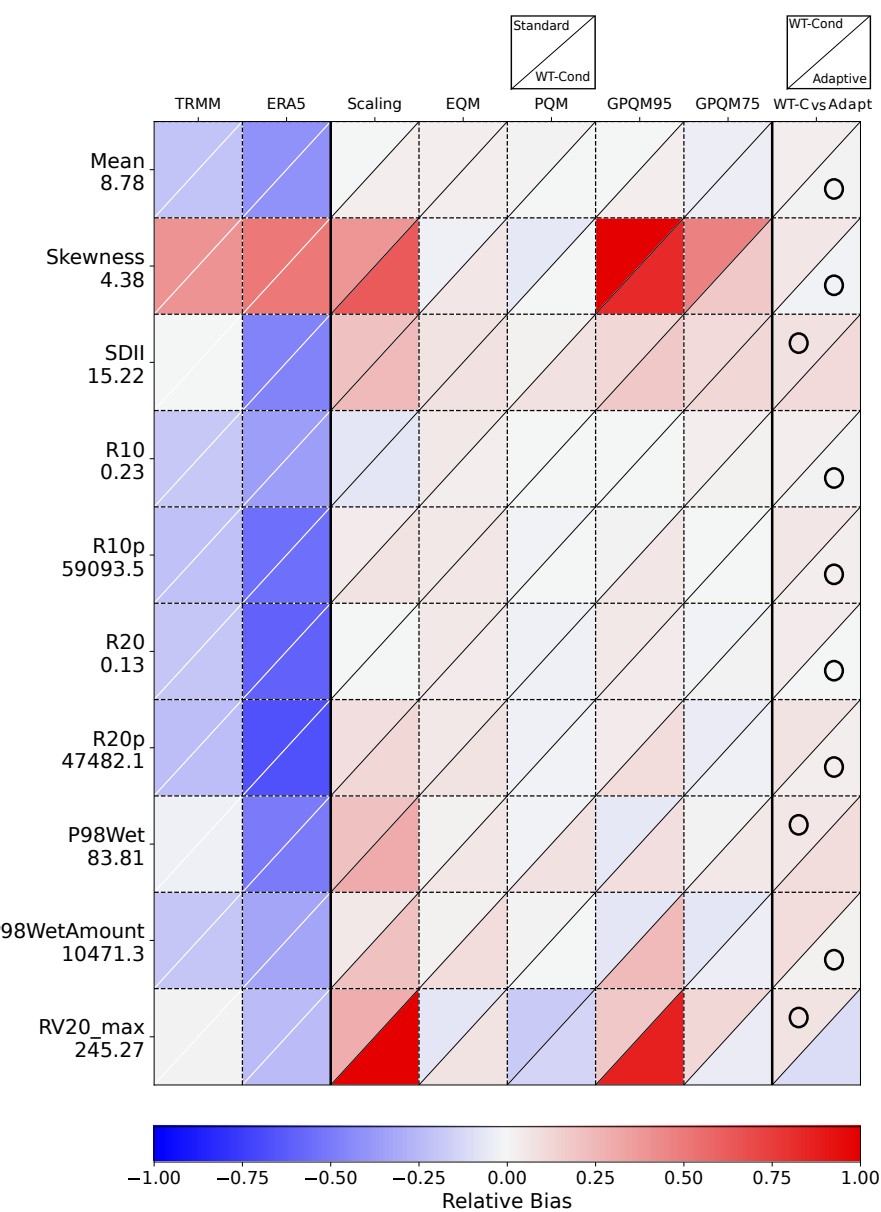

**Figure B6.** Same as Fig. B1, but for the Nuu'uli rain gauge location.

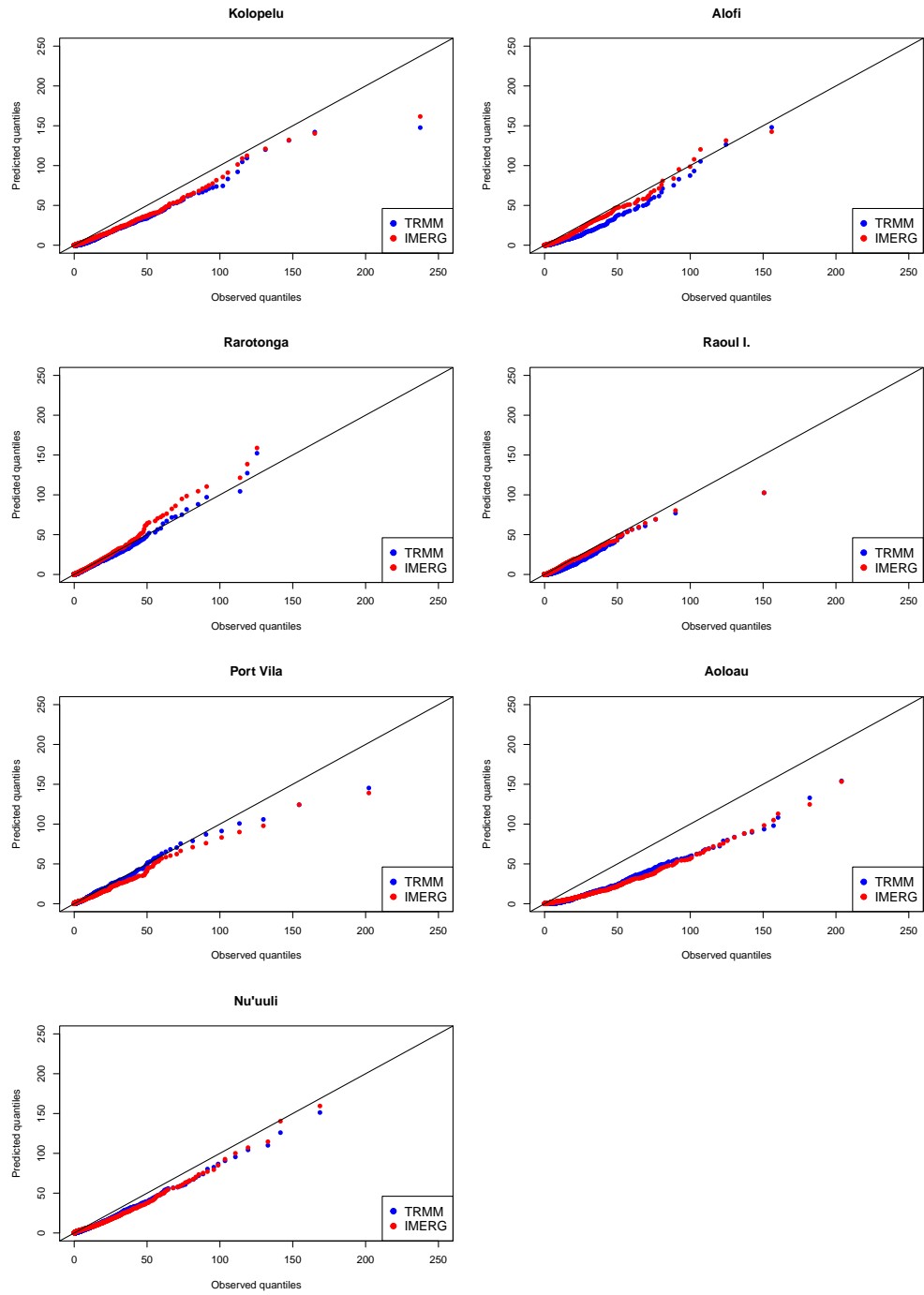

**Figure C1.** Quantile-quantile plots of TRMM (blue), and IMERG (red), against the rain gauge daily precipitation records of the stations indicated.

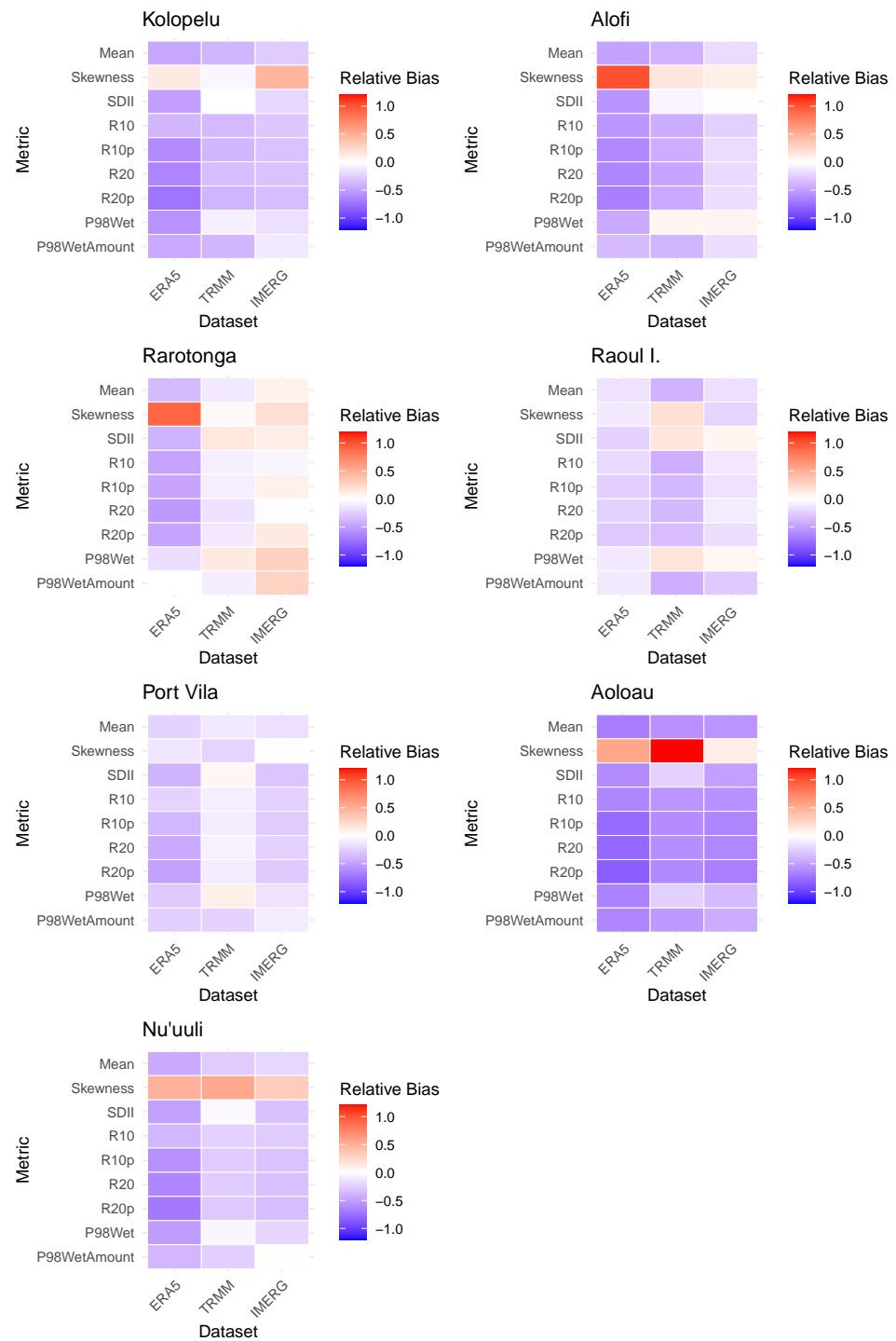

**Figure C2.** Relative Biases of the climate indices used for validation (Table 2 of raw TRMM and IMERG datasets, at the closest grid points of the locations of each reference station used in the study (Table 1). As an additional reference, we include in the first column the biases of the ERA5 reanalysis (Hersbach et al., 2020) raw precipitation.