# Peer review of "Refining Remote Sensing precipitation Datasets in the South Pacific with An Adaptive Multi-Method Calibration Approach"

_EGUsphere, 2023_

## Author Comment (AC1)

Response to comment <https://doi.org/10.5194/egusphere-2023-1402-RC1>, in open discussion of the manuscript entitled:

Refining Remote Sensing precipitation Datasets in the South Pacific: An Adaptive Multi-Method Approach for Calibrating the TRMM Product by Mirones et al. 2023

https://egusphere.copernicus.org/preprints/2023/egusphere-2023-1402/#discussion

*We thank the reviewer for his/her useful comments. We next provide a detailed response to all of them, and indicate the corresponding changes in the new revised version of the manuscript by means of the attached 'diff' file.*

**About the generalization of the method:**

*This study does not intend to extrapolate the proposed calibration method across the entirety of the low-middle latitudes, a point not claimed at any moment within our manuscript. The calibration method, as articulated, is inherently not compatible with a broad, generalized application. It relies on a weather type classification designed for the synoptic characterization of regional precipitation patterns. We have emphasized this characteristic of the method in the revised version of our manuscript (L61-64). Consequently, its viability is contingent upon a precedent weather type classification, signifying a predominantly local or regional scope rather than a hemispheric or global one. At the local and regional levels, our findings demonstrate a consistently low variability in the calibration results across the station set locations. This consistency serves to mitigate uncertainty in the calibrated data, with the adaptive nature of the approach tailored to different weather types corresponding to specific areas or regions. In this particular study, we introduce the TRMM calibration method focused on the South Pacific, within the area of influence of the South Pacific Convergence Zone (SPCZ). Consequently, a specific weather typing analysis was undertaken, accounting for the distinctive features of this region's precipitation seasonality, notably the impact of the SPCZ pulses (Australian Bureau of Meteorology and CSIRO 2011). Our classification takes into consideration the unique influence of tropical cyclones under specific conditions, effectively captured through our classification system. Comprehensive details on this classification can be found in our prior work (Mirones et al. 2022), presented separately to avoid unnecessary lengthening of the current manuscript.*

*Although this study does not aim for a direct generalization since a specific weather typing is first required–, the methodology presented provides the flexibility to customize the calibration process based on user preferences, including the available implementations of the different statistical correction methods, thereby allowing for specific indices, such as extreme rainfall indicators, to be assigned higher weights. Additionally, this capability enables effective calibration to address the impact of intense rainfall events on the overall distribution and*

*tailor the calibration to the specific bias aimed to be mitigated in the product. Furthermore, the proposed adaptive method is highly versatile and can be applied to different scenarios, datasets, and regions, provided that a prior weather typing classification exists to capture the pertinent processes related to regional precipitation patterns.*

***About the limited number of available rain gauge observations:***

*We acknowledge the critique regarding the limited number of rain-gauge observations employed in our study and concur with the referee on the necessity for additional clarification on this matter. We have included some additional details in the revised manuscript (Sec. 2.1).To address this concern, we have utilized the primary regional database available, namely the PACRAIN Database. This database comprises daily (and monthly) rainfall records from numerous sites situated on atolls and islands throughout the South Pacific. Despite the seemingly ample raw samples within the database, a meticulous examination of missing data and time series homogeneity reveals a significantly reduced number of suitable data points. Two critical considerations arise in this context:*

1. ***Bias Correction Requirement:*** *Achieving robust fits for the various statistical methods employed in bias correction demands a relatively large sample size. This is especially true for effectively characterizing extreme events, a main point in our study due to their paramount importance in numerous hydrological applications.*
2. ***Representativity of Locations:*** *The chosen locations must encompass a representative spectrum of variability within the region. The most relevant features of this region are mainly manifested as three extensive bands of large-scale wind convergence and associated rainfall. The convergence leads to the formation of the West Pacific Warm Pool (WPWP), partially overlapping with the study area in its northwestern part (Port Vila station, see Table 1). The Intertropical Convergence Zone (ITCZ), a precipitation band just north of the equatorial belt, is particularly robust between June and August (Waliser et al. 1993). In contrast, the SPCZ, a high-precipitation band between the Solomon Islands and the Cook Islands, is strongest between December and February (Vincent 1995), showing a significant temporal correlation with extreme precipitation events (Griffiths et al. 2003). Most of the stations are arranged along this band (Kolopelu, Aoloau, Nu'uuli in the center, Alofi in the center south and Rarotonga in the southeast). In our study, these locations are strategically distributed across the entire domain, offering a sensible representation of diverse precipitation regimes. A more detailed analysis and discussion of this aspect can be found in our prior work (Mirones et al. 2022).*

*These factors collectively underscore our deliberate and considered approach in selecting and utilizing these rain-gauge data for this study, despite the challenges posed by the limitations of the available regional database. The scarcity of sufficient observational data poses a significant constraint in numerous studies within this environmentally sensitive region, like ours. On the other hand, this fact highlights the necessity of depending on satellite products such as TRMM and further reinforces the crucial role of effective calibration techniques for refining remote sensing estimates, particularly for applications with direct implications on the ground. In light of these challenges, we assert that the subset of seven*

selected rain gauge locations (detailed in Table 1 of the manuscript) has been thoughtfully chosen to align with the specific objectives of our study. Despite the limitations imposed by the constrained number of gauge locations, we contend that this subset makes a meaningful contribution to achieving our research goals.

Moreover, while it holds true that we obtain a significant enhancement in calibration at only five of the seven locations scrutinized, it is noteworthy that, in no instance, do we observe a degradation of calibration when employing the adaptive method. This observation, albeit approached with prudence given the limited number of locations, implies that the adaptive calibration method is advantageous with regard to single-technique or conventional weather-type-conditioned approaches, particularly regarding the potential to improve the extreme event representation of the calibrated series. In the worst-case scenario, it not only maintains the quality of calibration but also avoids any deterioration, thereby reaffirming its inherent robustness.

***Minor comment***

In relation with Figure 3, yes, the horizontal axes represent the RF scores (bounded from 0 to 1). We have modified the figure accordingly in the new revised version of the manuscript to indicate it explicitly in the graphs.

[revised manuscript text omitted]

---

## Author Comment (AC2)

https://egusphere.copernicus.org/preprints/2023/egusphere-2023-1402/

Response to comment RC2 (https://doi.org/10.5194/egusphere-2023-1402-RC2), in open discussion of the manuscript entitled:

Refining Remote Sensing precipitation Datasets in the South Pacific: An Adaptive Multi-Method Approach for Calibrating the TRMM Product by Mirones et al. 2023

*We would like to express our sincere gratitude to the reviewer for his/her time and effort in reviewing our manuscript. We appreciate the constructive feedback and the valuable suggestions for improving the quality and clarity of our work. We appreciate the reviewers' comments and have revised the manuscript accordingly. We provide a point-by-point response to all the issues raised below. We have also attached a "diff" file to this response to facilitate the revision process. We hope that the reviewer will find the revised version satisfactory and worthy of publication.*

**Major comments:**

1. **The TMPA product was used as a calibration object. The reasons for choosing TMPA should be provided. In particular, its successor. i.e., IMERG, has been released to the public and is better than TMPA in multiple aspects such as resolution, covering period, and quality. Moreover, the TMPA products have stopped updating. I am confused that this study selected TMPA instead of IMERG.**

   *We appreciate the reviewer's feedback and concur that this issue deserves a mention in the paper. The main objective of our study is essentially methodological, and therefore a great part of our analysis can be undertaken with different products, and it is therefore not an exclusive or ad-hoc methodology for the TRMM-TMPA product. Furthermore, in this case our interest is to evaluate the performance of different calibration methods for precipitation data, and TMPA is an adequate baseline as a challenging case to this aim, given its good overall performance but loss of accuracy in the representation of some critical extremes. However, we respectfully disagree with the referee's assessment that choosing this dataset "does not make much sense". We believe that this dataset is relevant and appropriate for our research question, which is the introduction of a novel methodology for calibration. In our opinion, just because it is not the newest and most accurate product does not mean that a study that uses it as a reference to analyze a new calibration technique does not make sense. It is precisely its known lack of accuracy*

*in representing certain precipitation events that makes it particularly interesting for this purpose, given that the need for its adjustment has been addressed in many previous studies, often in a suboptimal manner (see Mirones et al. 2023 for an overview, and some references therein, e.g.: Aghakouchak et al. 2009, Almazroui 2011). On the other hand, although we are aware of the improvements introduced in the new IMERG product, the study of its possible errors and limitations is beyond our scope in this article, although we hope that the methodological aspects that we introduce in this work can serve as a reference for other authors in the future to analyze in more depth the characteristics of this more recent product or others. Therefore, far from being inappropriate, it opens the door for new applications using different and improved datasets.*

*In response to the reviewer's comments, we have added a new paragraph in the introduction to explain why TRMM data is suitable for our study and how our calibration methodology can be applied to other satellite precipitation products. Furthermore, to avoid giving a misleading idea of exclusivity of the methodology for a specific product, we have removed the TRMM reference in the title (the term has been included in the keywords, though, to facilitate indexing). We hope that our modifications address the reviewer's concerns and demonstrate that our study is relevant, novel, and generalizable, regardless of the dataset choice.*

2. **Methodology: the method section is one of the important content, but the current descriptions are too concise. Based on five weather types, the paper proposed an "adaptive calibration approach" to take advantage of the complementary strengths for four methods and further improve the accuracy of the TMPA product. However, the weighting method was not described well, especially its standards and rules. In addition, could the weighting method used in this paper provide optimal weights? In other words, whether this weighting method can maximize the advantages of different error adjustment methods. In addition, the classification of weather types was not provided. I understand the authors gave the related literature, but this is not enough. A detailed classification scheme should be provided as it is an important component for the "adaptive calibration approach".**

*We thank the referee for this comment. In order not to be excessively long, in the initial manuscript we have omitted most of the methodology related to weather typing, citing a previous article where the same methodology is applied. However, we understand the reviewer's point of view and have added a new paragraph giving some key details about the clustering methodology used, the rationale for the choices made and its suitability for conditioned calibration. In addition, we include in the supplementary material (New dedicated Appendix A) a new additional figure in order to summarize the main clustering results. It also provides some additional information regarding the frequency of tropical cyclone occurrence, that will aid in the discussion of the results.*

[Figure]

Figure A1. (a): Daily time series classification of weather types for the extended 41-year period 1979-2020. Tropical Cyclone (TC) occurrences are represented by the white dots. (b): Absolute number of days affected by TCs by year and WT.

*We thank the referee for this insightful comment regarding the potential of the adaptive calibration approach for optimization. However, as we have already emphasized in the manuscript, there is no such a thing as an "optimal" tuning. In this study, we have selected a battery of validation indices suitable for the characterization of the most important marginal aspects of local precipitation relevant for hydrological/impact applications, but this is not comprehensive and may lack some other characteristics relevant in specific research contexts (e.g. temporal validation of annual/seasonal cycles, autocorrelation function, dry-wet or wet-wet transition probabilities, spell durations etc…). Furthermore, the importance given to each of these aspects (for example, giving priority to extreme events rather than mean precipitation) may also determine the calibration method choice, as we illustrate here. That is the main reason we have avoided the term "optimization" in the title and focused more on "refinement", in the sense that the end-user priorities may alter the final method combination choice. However, and following the reasoning of the reviewer, it is possible to use the validation framework to test for different validation measure sets and weighting schemes searching for a minimization of the overall error. Due to the possibility of combining so many measures of performance, we introduce in this study the RF score, that makes possible*

*the integration of their combined performance into one single descriptor. However, it would be possible to search for the minimization/maximization of one single error/performance measure using the same approach, and following the reference code included as supplementary material to this aim. In order to clarify these aspects we have introduced a major modification of the validation method section, providing further details on the methodology and its justification. We hope that with these changes the rationale behind the proposed validation framework is clearer. Regarding the specification of "standards", we have used a subset of precipitation indices used in the international initiative VALUE. The main advantage of using a set of indices already used in an international intercomparison project is that it allows for a consistent, objective, and shareable assessment. We have also emphasized this aspect and provided further details on this validation framework.*

3. **Results and discussion: The analysis has many subjective descriptions, which is not recommended for scientific writing. I suggest the analysis provides some specific metric values and strengthens in-depth interpretations on the results.**

   *Thanks for this comment. We have made an effort to remove all statements that may sound "subjective" and provide a more thorough explanation of the results based on the specific measures introduced in the study.*

**Other comments:**

1. **Lines 36-37: this conclusion is true only in some areas, not all regions. I suggest revising this sentence to avoid misleading the readers.**

   *Thanks for this point, it can certainly be misleading. We have clarified it in the sentence*

2. **The differences between the four methods (i.e., scaling, eQM, pQM, and gpQM) should be provided and discussed, especially for their advantages and limitations. Due to their respective advantages, the adaptive calibration approach, which could consider their advantages, is necessary. Meanwhile, the adaptive calibration approach is not the best in all cases, which is worth analyzing.**

   *We have included two new paragraphs in the Results and Discussion section of the revised version of the manuscript addressing the results obtained in relation to the specific properties of the methods tested. Advantages and limitations of the various methods tested are presented and illustrated with the results obtained at diverse study locations.*

3. **Weather typing is important in this study. So, which five types? The description lacks detail.**

   *We have expanded the explanation regarding the weather typing. Now it should be clearer all the methodological aspects regarding the weather typing and why finally 5*

*weather types have been used in this study. In addition, some supplementary material has been added to give additional context information, as previously detailed in response to major comment 2.*

4. **Lines 159-161: could the authors provide some literature to support this point?**

*Thanks for pointing to this sentence. We have rephrased the whole sentence and included some references supporting this claim.*

5. **Lines 161-162: can the authors provide some study results (e.g., specific metric values) to support this point?**

*Thanks. Yes, the text already indicates Fig. 1, that supports this finding. However, we have expanded the explanation to be more explicit and provide further detail.*

6. **Lines 166-167: what results?**

*The sentence makes reference to the results presented in Mirones et al 2023. We have rephrased and extended the sentence to be more explicit.*

7. **Why did the adaptive calibration method not improve the accuracy at the Rarotonga and Nu'uuli stations?**

*In the case of Rarotonga, this statement is not completely true, since the adaptive calibration method is able to improve the performance using the weighted RF score. This particular result emphasizes the fact that model assessment is sensitive to user choices, such as the set of indices and measures used for validation and, as in this case, the weights assigned to each of them. As a result, when more weight is assigned to the performance of extreme precipitation indices, the adaptive calibration approach performs better than the standard calibration (PQM). Also, when considering the standard unconditioned calibration method (one single technique for the entire calibration period), the best performing method under the unweighted validation scheme is PQM, while when using the weighted RF score scheme, favoring the good performance of extreme indicators, it is the GPQM95 method. Therefore, it is important to remark that under this validation framework, the ranking of calibration methods is sensitive to arbitrary decisions in i) the battery of validation indices used and ii) the weight assigned to each measure to compute the overall score. This is now explicitly explained in the revised manuscript.*

*In the case of Nu'uli station, the adaptive calibration did not improve the performance, although it also attained a similar high RF score (~0.85/0.75 unweighted/weighted RF scores, Fig. 2a-b). Compared to other locations in Figs C1-C5 (Appendix C) and Fig.1 (Kolopelu), Nu'uli has moderate raw TRMM biases that are effectively corrected by both empirical and parametric quantile mapping, as reflected in Fig. C6 (Appendix C). GPQM performs similarly in general, but fails in reproducing distributional skewness and*

*RV20_max in the case of the adaptive approach. This is in connection with the limitations of the GPQM method for higher percentile thresholds, due to limitations in the sample size, which are accentuated in the case of the adaptive approach, as commented in relation with previous minor Comment 2. Thus, the adaptive calibration does not offer any advantage over the single-technique approach in this particular case. However, as we stress in the article, it also does not degrade the performance, but matches it. In fact, the adaptive approach enhances most indices, except RV20_max, P98Wet and SDII, albeit slightly, as indicated in the right column of Fig. C6.*

**Citation**: https://doi.org/10.5194/egusphere-2023-1402-RC2

[revised manuscript text omitted]

---

## Referee Report (RR1)

This is my second review for this manuscript. Significant improvements have been made by the authors. Thank you very much. Here are some comments that I hope will help the author improve this manuscript.

Comments

1. Line 12: please delete "(WT)".

2. Lines 41-47: the explanation about the selection of the TMPA product is not convincing and misleads readers. The IMERG inherits all attributes of TMPA. In addition, the precipitation estimates of TMPA came from multiple satellites with different orbits and satellite sensors, the same as IMERG. I think the effectiveness and applicability of the proposed approach should be based on the data that is being updated and better, rather than the data that has stopped updating, which does not make much sense.

3. The reliability of the results is doubtful, as the total number of rain gauges is too small, especially for P95 (see Table A1). Appropriate discussion for this should be added.

4. It is not clear that the TMPA used in this study is calibrated against a standard satellite product or uncalibrated or native product.

5. Line 111: please provide the full name of PCA at the first appearance.

6. Lines 115: 'in order">>in order to.

7. Line 127: please provide the full name of EOF.

8. I suggest providing the flowchart of the method used in this study.

9. Lines 199-202: The description of the weighting method is very crude and vague. Please provide detailed weighting criteria or theoretical support for Table A2. In my opinion, the weighting method used in this study is highly arbitrary.

10. Why introduce ERA5 as a reference to compute the bias? It makes no sense for validation. If necessary, please provide appropriate reasons.

11. The unweighted RF means equal weight for each metric?

---

## Author Response (AR2)

===================================================================

**RESPONSE TO EDITOR**

===================================================================

Dear Editor,
We hereby include a detailed response to the referee, hoping to adequately attend his/her questions, some of them already discussed and addressed in the previous iteration. There seem to be two main concerns that we again try to clarify, and that are now emphasized and further elaborated in this third manuscript review. For your convenience, we next describe the two main concerns raised and a summary of our reply and steps followed to clarify these issues.

1. **Regarding the use of an older precipitation product**. We would like to address the referee's concern regarding our use of an older satellite product (TRMM) instead of the currently available IMERG database. To verify the referee's claims about the absence of biases in IMERG, we conducted an additional analysis. While IMERG indeed inherits many attributes from TRMM, including the use of multiple satellites and sensors, and offers improved accuracy and geographical coverage, our analysis indicates that the biases observed in TRMM persist in the newer IMERG products. Given that these biases remain at the locations of our study, we believe that the choice of data source is of secondary importance for illustrating our calibration methodology. The primary focus of our study is on the nature and application of the calibration methodology itself. This approach not only highlights the importance of calibration but also demonstrates the applicability of our methods to both older and newer datasets. Therefore, we consider both the calibration methodology and the results obtained to be perfectly valid and relevant. Additionally, our methods are easily applicable to new datasets, as we have included an open-source notebook to facilitate the reproducibility of the results.

2. **Regarding the limited number of calibration sites**. As we indicated in the manuscript, this limitation is due to the availability of high-quality records over a sufficiently long temporal period, which is essential to ensure the robustness of calibration. Our study focuses on an area where a previously established weather typing exists, a necessary condition for the application of the methodology. This weather typing is crucial for describing the seasonality and other precipitation patterns in the region, providing a relevant context for our analysis, in which calibration is conditioned to specific situations. Despite the limited number of stations, our results are consistent and clearly demonstrate the validity and usefulness of the adaptive calibration approach. The consistency of our findings across the available stations supports the reliability of our methodology. Additionally, the high-quality, long-term data we used enhances the credibility of our results. We believe that the primary objective of our research is methodological. Therefore, it is not essential to include a large number of locations. Instead, we aim to demonstrate the methodology's application and its performance across well-scattered locations within the weather typing domain, each with

varying conditions. Following the referee's concern about lack of robustness of the results, it is important to note that the indicated P95 in manuscript's Table B1, does not depend on the number of stations but on the length of the time series, which is sufficient to obtain robust results, as already explained in the manuscript. This is precisely why we selected a reduced set of locations, applying a stringent criterion to ensure long time series for a robust conditioned calibration.

We hope this clarifies our position and addresses the referee's concerns. With this thorough revision, we sincerely hope that our study meets the high-quality standards of the HESS Journal and will be accepted for publication.

Yours sincerely,

J. Bedia,
On behalf of the authors

===================================================================

**RESPONSE TO REFEREE**

===================================================================

*This is my second review for this manuscript. Significant improvements have been made by the authors. Thank you very much. Here are some comments that I hope will help the author improve this manuscript.*

Thank you very much for your positive feedback. We have made significant efforts to enhance the original manuscript, incorporating the valuable suggestions provided by the referee in his/her first review. We also appreciate the referee's new comments, which we believe will help to better communicate the significance of our study and the relevance of its main findings.

Throughout our response, we will interweave the referee's comments (highlighted in italics) with our replies.

We thank the referee again for their time and positive feedback, and sincerely hope that our responses will meet his/her expectations.

**Comments**

1. Line 12: please delete "(WT)".

Done.

*2. Lines 41-47: the explanation about the selection of the TMPA product is not convincing and misleads readers. The IMERG inherits all attributes of TMPA. In addition, the precipitation estimates of TMPA came from multiple satellites with different orbits and satellite sensors, the same as IMERG. I think the effectiveness and applicability of the proposed approach should be based on the data that is being updated and better, rather than the data that has stopped updating, which does not make much sense.*

Thank you for your insightful comments regarding the selection of the TMPA product. We understand your concerns about using data that is no longer updated. However, we included the TMPA data specifically to illustrate the strong biases present in older datasets, which effectively highlight the impact of different calibration methods.

While it is true that IMERG inherits many attributes from TMPA including the use of multiple satellites and sensors, and improves its overall representation of precipitation, our analysis shows that the biases observed in TMPA are still present in the newer IMERG products. Our analysis shows that the biases observed in TMPA are still present in the newer IMERG products at the selected localities. In Fig.1, we depict the biases of both TMPA and IMERG using a quantile-quantile plot against the rain gauge records. In Fig. 2, these biases are more accurately quantified in terms of the different validation indices used in this study (Table 2 of the manuscript). The new IMERG product shows a moderate overall improvement over the old TMPA product in some sites (Alofi, Raoul Island); however, biases are still present, and in some locations are even higher that TMPA (Port Vila) and therefore also require calibration against a 'ground-truth' reference.

Therefore, for the purpose of illustrating our calibration methodology, the choice of data source is, in our view, of secondary importance, as the emphasis of our study is on the nature and application of the calibration methodology itself. We believe this approach not only underscores the importance of calibration but also validates the applicability of our methods to both older and newer datasets.

We appreciate your feedback and hope this explanation clarifies our rationale for including the TMPA data. We have included a mention to this rationale in the new revised version of our manuscript (Lines 43-49), and also included the figures above in the new Appendix C to showcase the remaining biases in the IMERG dataset.

[Figure]

Figure 1. Quantile-quantile plots of TRMM (blue), and IMERG (red), against the rain gauge daily precipitation records of the stations indicated.

[Figure]

Figure 2. Relative Biases of the climate indices used for validation (Table 2 of the manuscript) of raw TRMM and IMERG datasets, at the closest grid points of the locations of each reference station used in the study (Table 1 of the manuscript).

*3. The reliability of the results is doubtful, as the total number of rain gauges is too small, especially for P95 (see Table A1). Appropriate discussion for this should be Added.*

Thank you for your valuable feedback regarding the number of rain gauges used in our study. We acknowledge that the total number of gauges is limited. However, as we already indicate in the manuscript, this limitation is due to the availability of high-quality records over a sufficiently long temporal period, which is essential to ensure the robustness of our calibration results.

Furthermore, our study is focused on an area where a previously weather typing exists (Mirones et al. 2023). This weather typing is crucial for describing the seasonality and other precipitation patterns in the region, providing a relevant context for our analysis, in which calibration is conditioned to specific situations.

Despite the limited number of stations, our results are consistent and clearly demonstrate the validity and usefulness of the adaptive calibration approach. The consistency of our findings across the available stations supports the reliability of our methodology. Additionally, the high-quality, long-term data we used enhances the credibility of our results. We believe that the primary objective of our research is methodological. Therefore, it is not essential to include a large number of locations. Instead, we aim to demonstrate the methodology's applications and its performance across a few locations, well scattered across the weather typing domain, with varying conditions. Following the referee's comment, please note that the indicated P95 in the above-mentioned Table, on the other hand, does not depend on the number of stations, but on the length of the time series, one of the reasons for choosing such a reduced set of locations, containing a sufficient amount of data for a conditioned calibration.

We believe that our study provides a valuable contribution to the field, illustrating the effectiveness of our calibration approach even with a limited number of stations. We appreciate your consideration of these points and hope this explanation addresses your concerns. The points above addressed are already indicated in the manuscript (Sec. 2.1)

4. It is not clear that the TMPA used in this study is calibrated against a standard satellite product or uncalibrated or native product.

Thank you for pointing out the need for clarity regarding the TMPA product used in this study. The TMPA product used is the 3B42_7B version (as already outlined in Sec. 2.1), a research-grade dataset designed for climatological and hydrological analyses. This product is calibrated by combining satellite-based precipitation estimates with ground-based gauge data to enhance accuracy. Specifically, it applies bias corrections using monthly gauge analysis from the Global Precipitation Climatology Centre (GPCC), which helps reduce systematic errors present in the raw satellite estimates. Unlike the real-time 3B42RT product, the 3B42_7B version undergoes additional processing to ensure consistency and reliability, making it a suitable choice for retrospective scientific studies.

Despite these calibrations, the product still exhibits significant biases, as demonstrated in our study. We have included extended information in the revised text (Sec. 2.1, Lines 107-115) to include further technical details of this TRMM version.

5. Line 111: please provide the full name of PCA at the first appearance.

Done

6. Lines 115: 'in order">>in order to.

Done

7. Line 127: please provide the full name of EOF.

Done

8. I suggest providing the flowchart of the method used in this study.

Thanks for this suggestion aimed at improving clarity. While we acknowledge that our approach is methodologically more complex than a direct calibration, we believe it is not so intricate as to necessitate a flowchart representation. We are committed to making our study as clear and comprehensive as possible, and we believe that the detailed descriptions and the prepared data and code we have included already ensure the reproducibility of each step in our study. However, we are open to further discussion if you believe that a specific aspect of our method could benefit from additional visual representation.

9. Lines 199-202: The description of the weighting method is very crude and vague. Please provide detailed weighting criteria or theoretical support for Table A2. In my opinion, the weighting method used in this study is highly arbitrary.

Firstly, we are unsure what is meant by "crude." Regarding the term "vague," we believe our method is quite detailed and addresses key aspects of precipitation necessary for an adequate assessment of the calibrated series. Our approach leverages well-defined validation indices that consider various statistical aspects of precipitation (e.g., frequency, mean amount, extremes, overall distribution shape) and associated performance measures (e.g., biases—relative and/or absolute—and correlation). These are clearly indicated in Table 2 and supported by the references listed in our manuscript.

Moreover, we believe the validation performed in our study surpasses the typical evaluations conducted in the calibration of satellite products, which often focus on simpler indicators like monthly accumulations. This is also highlighted in the references of our manuscript.

We have already introduced several clarifications in response to your initial comments. It is important to emphasize that the validation of a calibration method is inherently multi-faceted, with results varying based on the aspects considered. Therefore, we have adopted a rigorous approach to ensure a comprehensive validation, following the VALUE framework, which is widely used in the Euro-CORDEX community's coordinated downscaling activities.

Given the multivariate nature of the validation, our weighting method provides users with maximum flexibility to incorporate various metrics into the overall evaluation. This method allows for the assignment of greater weight to certain aspects over others, depending on the context (e.g., emphasizing extremes for hydrological studies or frequencies for agricultural studies). Calibration can alter these distributional aspects of precipitation, necessitating a comprehensive evaluation method.

The introduction of the ranking framework (RF) enables us to rank different calibration methods by summarizing individual results for each validation measure into a single metric. This approach is not new and has been used in previous model validation work, such as the observational uncertainty analysis in Euro-CORDEX (Kotlarsky et al., 2019). We have adopted this robust and widely accepted methodology in our study.

While there is some degree of arbitrariness in selecting validation metrics and assigning weights, we view this as a strength rather than a weakness. This flexibility allows the method to be tailored to different needs. The specific set of validation indices and weights used in our study serves as an example that users can modify as needed, following the reproducibility examples provided.

We hope this clarifies our approach and addresses your concerns.

10. Why introduce ERA5 as a reference to compute the bias? It makes no sense for validation. If necessary, please provide appropriate reasons.

Thank you for raising the issue of using ERA5 as an additional reference to compute biases. We understand your concern about its role in validation. However, please note that the use of ERA5 here is purely comparative, and it is not used elsewhere in the calibration process. In any case, we think that the inclusion of these data adds value to the results presented, helping to better contextualize the magnitudes of the biases found. We deemed it relevant to use ERA5 as a reference to compare its biases with satellite data for several reasons:

1.  High-Quality Data: ERA5 is a state-of-the-art reanalysis product that integrates a vast array of observations from satellites, ground stations, and other sources using advanced data assimilation techniques. This makes ERA5 data high-quality, consistent, and reliable, widely used by the community in many impact applications, despite some remaining biases.
2.  Detailed Resolution: ERA5 offers high spatial (~31 km) and temporal (hourly) resolution, which is crucial for capturing detailed precipitation patterns. This comprehensive

coverage allows for a more meaningful comparison with satellite data across different regions and time periods.

3. Proven Accuracy: Studies have shown that ERA5 has smaller biases compared to its predecessors and other reanalysis products. For example, ERA5 exhibits lower bias in tropical regions compared to ERA-Interim, MERRA-2 and JRA-55 (Hassler and Lauer 2021), and improves global-mean correlation of precipitation with monthly-mean GPCP, making it a more accurate reference for precipitation data (Hersbach et al. 2020).

4. Consistency with Previous Work: We used ERA5 for a previous weather typing (Mirones et al. 2023), which we later applied to condition our satellite precipitation calibration. Presenting the biases of ERA5 in this context provides continuity and consistency with our earlier work, further validating our approach.

By using ERA5 as a reference, we aim to provide a high-quality benchmark to contextualize the biases of satellite data. This comparison helps to highlight the strengths of the satellite products while also underscoring the importance of calibration against a reliable reference, as satellite product biases remain an issue.

We agree with the referee that providing a rationale for this choice would enhance the overall quality of our manuscript. We have succinctly addressed this in L231.

11. The unweighted RF means equal weight for each metric?

Yes, we have included a sentence to clarify this term when it first appears in the text.

**References**

Hassler, B., Lauer, A., 2021. Comparison of Reanalysis and Observational Precipitation Datasets Including ERA5 and WFDE5. Atmosphere 12, 1462. https://doi.org/10.3390/atmos12111462

Hersbach, H., Bell, B., Berrisford, P., Hirahara, S., Horányi, A., Muñoz-Sabater, J., Nicolas, J., Peubey, C., Radu, R., Schepers, D., Simmons, A., Soci, C., Abdalla, S., Abellan, X., Balsamo, G., Bechtold, P., Biavati, G., Bidlot, J., Bonavita, M., Chiara, G., Dahlgren, P., Dee, D., Diamantakis, M., Dragani, R., Flemming, J., Forbes, R., Fuentes, M., Geer, A., Haimberger, L., Healy, S., Hogan, R.J., Hólm, E., Janisková, M., Keeley, S., Laloyaux, P., Lopez, P., Lupu, C., Radnoti, G., Rosnay, P., Rozum, I., Vamborg, F., Villaume, S., Thépaut, J., 2020. The ERA5 global reanalysis. Q.J.R. Meteorol. Soc. 146, 1999–2049. https://doi.org/10.1002/qj.3803

Kotlarski, S., Szabó, P., Herrera, S., Räty, O., Keuler, K., Soares, P.M., Cardoso, R.M., Bosshard, T., Pagé, C., Boberg, F., Gutiérrez, J.M., Isotta, F.A., Jaczewski, A., Kreienkamp, F., Liniger, M.A., Lussana, C., Pianko-Kluczyńska, K., 2019. Observational uncertainty and regional climate model evaluation: A pan-European perspective. Int J Climatol 39, 3730–3749. https://doi.org/10.1002/joc.5249

Mirones, Ó., Bedia, J., Fernández-Granja, J.A., Herrera, S., Van Vloten, S.O., Pozo, A., Cagigal, L., Méndez, F.J., 2023. Weather-type-conditioned calibration of Tropical Rainfall Measuring Mission precipitation over the South Pacific Convergence Zone. International Journal of Climatology 43, 1193–1210. https://doi.org/10.1002/joc.7905

---

## Author Response (AR3)

**In terms of the weighting method, the authors did not answer my previous question correctly. I want to know why the weight values of some metrics (e.g., Mean, SDII) are 0.05, and some are 0.15, 0.1, or 0.2. Why not are 0.01, 0.02, 0.03, ...(Just an example, not meaning that this weighting method is correct)? This requires robust theoretical support.**

Dear referee,

Thank you again for your insightful comments to our paper, and also your question about the RF score weighting methodology. We apologize for any confusion caused by our previous response.

To clarify, the weights assigned to each metric in our validation methodology are designed to sum to one, ensuring a balanced and normalized approach, as indicated in eq. 3 in our manuscript:

$$RF_j = \frac{1}{N} \sum_{i=1}^{N} w_i \cdot Z'_{i,j} \quad \text{where} \quad \sum_{i=1}^{N} w_i = 1$$

Where RF is the "ranking framework" score for calibration $j$, $N$ is the number of validation indices considered in the validation procedure, $Z'_{i,j}$ is the (absolute) normalized bias for index $i$ and calibration $j$ and $w_i$ is the weight assigned to each validation index. In our manuscript we have considered a set of N=10 indices widely used to describe different aspects of precipitation, and also some arbitrary weights in the case of the weighted RF scheme, for illustration purposes.

Normalization of weights is a straightforward method where we assign weights based on criteria such as the importance or impact of the validation measure within the validation procedure. For example, we might assign a higher value to performance in extreme indices. Each weight is then divided by the total sum of all weights to ensure they sum to one. This method is often used in multi-criteria decision analysis, providing a clear and intuitive approach for weighted validation schemes.

The specific values (e.g., 0.05, 0.15, 0.1, 0.2) were chosen to illustrate the potential for varying weights based on different criteria. These values are illustrative and do not serve any specific purpose beyond demonstrating the possibilities of the weighted approach. We acknowledge that an equally weighted scheme is often a suitable alternative, particularly when aiming to avoid arbitrary choices.

For instance, the weight of 0.05 for the SDII bias reflects its relative importance in our analysis. Biases in this index are less penalized than biases in other indices with higher weights, such as 0.2 for the P98 wet amount. The value of 0.05 indicates that the performance of the calibration method in this specific index accounts for 5% of the total RF score, which is appropriate for a calibration procedure that does not prioritize this indicator highly. Similarly, the weights of 0.15, 0.1, and 0.2 for other metrics were chosen to reflect their respective contributions to the overall validation process.

We have introduced an explanatory sentence in the new revised version of our manuscript (Sec. 2.4.2) to better explain this point. We hope this explanation provides the robust theoretical support you requested. Please let us know if further clarification is needed.